# Lightweight Latent Verifiers for Efficient Meta-Generation Strategies

## Abstract

Verifiers are auxiliary models that assess the correctness of outputs generated by base large language models (LLMs). They play a crucial role in many strategies for solving reasoning-intensive problems with LLMs. Typically, verifiers are LLMs themselves, often as large (or larger) than the base model they support, making them computationally expensive. In this work, we introduce a novel lightweight verification approach, LiLaVe, which reliably extracts correctness signals from the *hidden states* of the base LLM. A key advantage of LiLaVe is its ability to operate with only a small fraction of the computational budget required by traditional LLM-based verifiers. To demonstrate its practicality, we couple LiLaVe with popular meta-generation strategies, like best-of-$n$ or self-consistency. Moreover, we design novel LiLaVe-based approaches, like conditional self-correction or conditional majority voting, that significantly improve both accuracy and efficiency in generation tasks with smaller LLMs. Our work demonstrates the fruitfulness of extracting latent information from the hidden states of LLMs, and opens the door to scalable and resource-efficient solutions for reasoning-intensive applications.

## 1 Introduction

Recently, there has been substantial interest in enhancing the *reasoning capabilities* of large language models (LLMs). Specifically, this effort includes applying LLMs to solve mathematical problems (Cobbe et al., 2021; Trinh et al., 2024), writing code (Jiang et al., 2025), automating scientific discovery (Novikov et al., 2025), recognizing complex spatial patterns (Chollet et al., 2024), and writing formal proofs (Mikuła et al., 2024; Lin et al., 2025).

Efforts to improve LLM performance on reasoning-intensive tasks have followed two primary directions. *First*, there is a substantial body of work focusing on *pre-training* or *fine-tuning* models targeting reasoning-intensive tasks. To this end, high-quality, reasoning-focused data are collected, like OpenWebMath (Paster et al., 2023), or Proof Pile (Azerbayev et al., 2023). In addition to that, new training methodologies are being developed, such as self-improvement loops (Zelikman et al., 2022), or reinforcement-learning-based approaches (Guo et al., 2025; Zhang et al., 2025), which emerged as remarkably effective in the context of reasoning tasks.

*Second*, there is ongoing research into designing *inference-time techniques* to enhance the performance of LLMs on reasoning-focused tasks, where an LLM is already pre-trained and fixed (Welleck et al., 2024). The examples of two such simple, yet effective techniques are *chain-of-thought* prompting (Wei et al., 2022), and *self-consistency decoding* (Wang et al., 2023), also known as *majority voting*. More advanced inference-time approaches often combine decoding process from the base LLM with a *verifier* (often also called a *reward model*), trained to assess the correctness of individual reasoning steps – or entire reasoning trajectories – in order to enhance the base model's performance (Cobbe et al., 2021; Lightman et al., 2023). Typically, such verifiers are LLMs themselves, often as large – or larger – than the base model they support, making them computationally expensive. For instance, in a recent work by Wu et al. (2025), an LLM verifier of size 34B of parameters is paired with base models of size 7B and 34B parameters.

The increased inference-time computational cost resulting from using large, LLM-based verifiers may be a significant limitation. This is especially important in setups where the verifier is called multiple times for decoding one sample, which is the case in verifier-guided tree search approaches like in the REBASE algorithm from Wu et al. (2025). Moreover, large verifiers may be costly to train.

This is especially important given the indication from the literature (Havrilla et al., 2024; Wang et al., 2024a) that the performance of LLM-based verifiers does not transfer across different base LLMs.

Our work aims to introduce *computationally efficient* verifiers (both in *training* and *inference*), which can be used to enhance the performance of the base LLMs in reasoning-intensive tasks. To this end, we develop LiLaVe – Lightweight Latent Verifier, which is a simple and practical method for extracting the correctness signal from the *hidden states* of the base LLM (Section 2). LiLaVe is based on a fast, classical machine learning model – gradient boosted decision trees. It can be trained quickly ($< 15$ min) on CPU only using a small number of LLM samples ($\approx 5$k), to reliably extract the hidden correctness signal. This makes our approach easily adaptable to new datasets and models.

In Section 3, through a series of experiments, we demonstrate how LiLaVe can be practically and effectively used to implement various *meta-generation strategies* focused both on *correctness of the inferred answers* as well as on *inference-time compute-efficiency*.

Besides the practical advantages of our approach, it reveals an intriguing phenomenon: hidden states of LLMs carry useful information which can be uncovered by classical machine learning methods.

In summary, our contributions are as follows:

- We introduce LiLaVe: a novel lightweight verification approach that extracts correctness signals from the hidden states of the base LLM; we show that in terms of the AUC metric it outperforms other lightweight approaches and is competitive with compute-intensive LLM-based verifiers.

- We experimentally study which hidden states across the model's layers and the output tokens provide the optimal correctness signal, which brings introspection into the model's mechanics.

- Finally, we show how LiLaVe brings practical advantage by coupling it with several meta-generation strategies and demonstrating significantly improved accuracy and inference-time compute-efficiency of LLMs on reasoning tasks compared to the baselines. In particular:
  - We introduce the *conditional majority voting* approach, which reduces the average inference cost while maintaining high accuracy.
  - We demonstrate the effectiveness of the *conditional self-correction* approach, in which the base model is asked to self-correct only when the LiLaVe verifier's score is low.

## 2 METHOD

Our approach to improving the accuracy and efficiency of LLMs on reasoning-intensive tasks at test time involves two key components. First, we train a lightweight latent verifier (LiLaVe) using selected hidden states extracted from the LLM during the generation of CoT-style solutions of mathematical problems, labeled by the correctness of the final answers concluding them (see Section 2.1). Subsequently, we employ the verifier to estimate the probability of LLM's answers being correct and integrate it with various *meta-generation strategies* (described in Section 2.2).

### 2.1 LIGHTWEIGHT LATENT VERIFIER – LILAVE

**Data** Given a question $q$, an LLM generates an answer sequentially as $y = y_1 y_2 \cdots y_m$, where $y_i$s are individual tokens. During the decoding, we extract hidden states $h_t^l \in \mathbb{R}^n$ representing the activations from the $l$-th transformer's layer at the generation of the $t$-th token, where $n$ is the hidden dimension of the model. Instead of extracting hidden states from all possible locations $(t, l)$, we fix sets of indices $L, T$ and require $l, t \in L \times T$. In Sec. 3.2 we experimentally determine optimal $L, T$.

While the answer $y$ contains the chain-of-thought style reasoning, we determine its correctness solely by looking at the final answer. To evaluate correctness, we use an automated evaluator that compares the generated final answer to the ground truth, resulting in a binary correctness label $c$. Finally, a dataset $\mathcal{D}$ for training LiLaVe consists of datapoints of the form of quadruples $(h_t^l, l, t, c)$. Note that we extract $|L| \cdot |T|$ hidden states per one generation. Therefore, if $Q$ is the dataset of questions and we sample $k$ generations for each $q \in Q$, we have $|\mathcal{D}| = |L| \cdot |T| \cdot |Q| \cdot k$.

**Training** Having collected $\mathcal{D}$, we train an efficient classifier $M$ to predict the binary label $c$ given the hidden state $h_t^l$ and its location given by the indices $l, t$. The output score $M(h_t^l, l, t) \in [0, 1]$ is to be interpreted as the probability of the response $y$ to be correct.

We tested several classifiers suitable for such data: logistic regression (Hastie et al., 2009), SwiGLU (Shazeer, 2020), and gradient boosted trees (Friedman, 2001). In our initial experiments, we observed that the last method (concretely, its XGBoost implementation by Chen & Guestrin (2016)) performed best and most robustly (see Appendix B.2). Therefore, we chose to rely on this classifier.

**Inference** During inference, the base language model generates a response $y$ along with a set of associated hidden states $H_y$, which are indexed by their locations $(l, t)$. We then apply the trained XGBoost model $M$ to predict a score $s_h$ for each hidden state $h \in H_y$. Finally, these scores are aggregated, which results in the final correctness estimate, *i.e.*, the LiLaVe score:

$$\text{LiLaVe}(y) = \text{aggregate}(\{s_h\}_{h \in H_y}) \in [0, 1].$$

After experimenting with several aggregation methods – taking minimum, maximum, or average score – we chose to use averaging as it performed best.

## 2.2 LiLaVe-based meta-generation strategies

We consider several *meta-generation strategies*, i.e., strategies that build on top of the *base generator* (the base language model) and a trained LiLaVe verifier. First, we experiment with two standard approaches: **best-of-$n$** sampling and **weighted majority voting**. In both approaches, we first sample $n$ responses from the base generator with fixed temperature $t > 0$. As the final response in best-of-$n$, we select the one with the highest LiLaVe score. In weighted majority voting, we perform a majority voting across the final answers extracted from $n$ full responses, weighted by their LiLaVe scores.

Standard majority voting and its weighted variant are effective techniques; however, they may be computationally expensive as they require generating multiple independent samples per question. In the weighted voting, there is an additional cost of extracting hidden states from the decoded samples, which may cause a significant slowdown in practical settings.

This motivates our novel approach of **conditional majority voting**: first, we generate a single sample from the base generator, and we score it with LiLaVe. If the score is above a predetermined threshold $s \in [0, 1]$, we consider the sampled response as final. Otherwise, we interpret the low score as an indication of the base model's mistake or uncertainty, and generate $n$ additional samples to perform a majority voting to determine the final response.

Finally, we investigate another new meta-generation strategy of **conditional self-correction**. Prompting LLMs to verify and correct their responses gives varied results (Huang et al., 2024). LLMs, indeed, often are able to fix their mistakes, but at the same time, they tend to turn correct responses into incorrect ones. This makes the self-correction procedure unreliable and, in most cases, overall unsuccessful. In the *conditional* self-correction, we leverage LiLaVe to achieve reliable accuracy improvements. First, we generate the initial response and score it with LiLaVe. Then, we prompt the model to self-correct its response[1] only if the LiLaVe score is below a predetermined threshold $s$.

In Section 3.4, we demonstrate the performance of these LiLaVe-based meta-generation strategies on several reasoning-intensive benchmarks.

## 3 Experiments

In this section, we describe the experiments we conducted in order to develop and evaluate LiLaVe. First, in Section 3.1, we describe four reasoning-intensive, mathematical benchmarks that we used. In Section 3.2, we study the influence of the location of extracted hidden states as well as sampling temperature on the predictive performance of LiLaVe. In Section 3.3, we introduce two alternative baseline methods for estimating the correctness of the LLM reasoning, which we subsequently compare with LiLaVe. Finally, in Section 3.4, we harness LiLaVe to four meta-generation strategies described in Section 2.2, and we demonstrate that despite being so lightweight, our verifier allows us to achieve substantial performance gains on the mathematical benchmarks.

Our experimental results demonstrate that LiLaVe excels in extracting the correctness signal from the internal states of the base LLM, and that this signal can be practically utilized in meta-generation strategies, improving the performance and efficiency on reasoning-intensive benchmarks.

---

[1]The specific self-correction prompt we use is shown in Figure 15 in Appendix C.

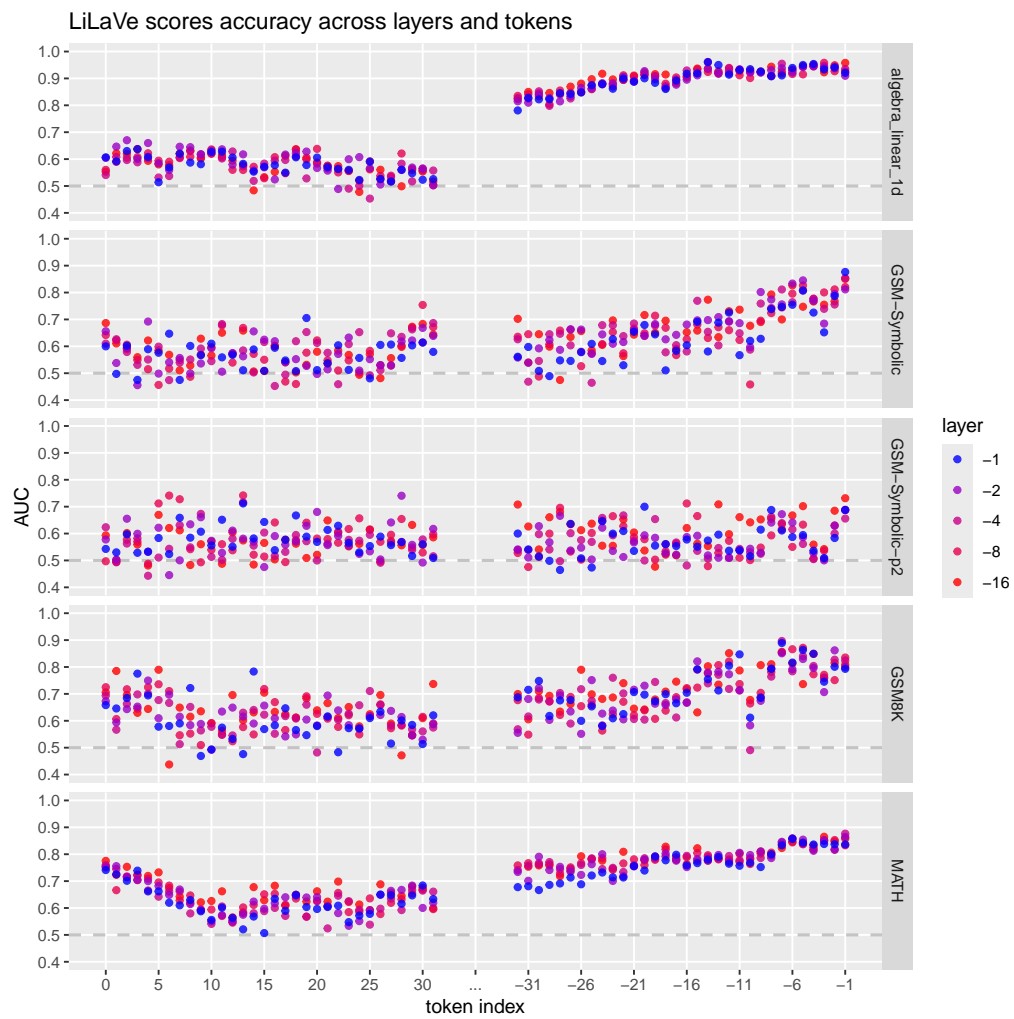

Figure 1: Predictive performance of LiLaVe on individual locations of hidden states determined by the indices of the transformer's layer and the sequence's token. We test the tokens from the prefix and suffix of the generated sequences, both of length 32. It is visible that the higher-quality signal can be retrieved from the final tokens; however, interestingly, even for the first tokens, LiLaVe provides a signal significantly better than the random baseline (dashed lines). At the same time, we cannot conclude which transformer layers give the best signal.

## 3.1 REASONING-FOCUSED BENCHMARKS

We evaluate LiLaVe and LiLaVe-based meta-generation strategies on four mathematical QA datasets, whose difficulty is appropriate for the LLMs we use: GSM8K (Cobbe et al., 2021), MATH (Hendrycks et al., 2021), GSM-Symbolic (Mirzadeh et al., 2024), and algebra_linear_1d (Saxton et al., 2019). The last two datasets are synthetic and therefore avoid potential contamination effects. For each of the benchmarks we select 1000 training examples to train a dataset-specific LiLaVe. We test on sets of 500–1319 examples, depending on the dataset. See Appendix A for more details regarding data.

## 3.2 DEVELOPING LiLaVe

Below, we describe experiments determining (1) the location of extracted internal language model information as well as (2) sampling temperatures resulting in optimal LiLaVe's performance.

In our main experimental line we use Llama 3.1 8B as the base model. To test the universality of LiLaVe, we also experimented with popular Gemma 2 2B and Phi-3.5-mini models – see Appendix B.

**Hidden states locations** As described in Section 2.1, we train LiLaVe on hidden states extracted from the base language model. The hidden states we extract correspond to different layers of the transformer model as well as different tokens in the decoded sequences. It is not clear which of those locations can allow for extracting the best correctness signal, therefore, we run an experiment aiming to answer this question. We fix a set of layer indices $L$ and token indices $T$ as:

$$L = \{-1, -2, -4, -8, -16\},$$
$$T = \{0, 1, 2, 3, \ldots, 31, -32, -31, \ldots, -3, -2, -1\}.$$

Negative indices follow the Python convention of list indexing: the element $-n$ is the $n$th element counting from the end of the list. For each $(l, t) \in L \times T$, we train a separate XGBoost model $M_{l,t}$ on hidden states corresponding to layer $l$ and token $t$. Then, we evaluate each of the trained models $M_{l,t}$ on a testing partition (using the corresponding hidden states), and calculate its predictive performance using the AUC metric.[2]

Figure 1 presents results of the experiment for the four datasets (introduced in Section 3.1; for GSM-Symbolic, we also consider its more difficult p2 variant).

First, we observe that, predictably, the correctness signal is better in the suffix of the decoded sequences (which is especially noticeable for algebra_linear_1d). However, curiously, the signal in the prefix of the decoded sequences is still significantly better than the random baseline (AUC = 0.5), which is especially visible for the first few tokens in the MATH dataset. Another observation is that there is no significant distinction between different transformer's layers, and even layers as deep as $-16$ provide good signal (Llama 3.1 8B used in this experiment has 32 layers in total.)

Based on the obtained results, we fix the following sets of indices of layers $L_{\text{LiLaVe}}$ and tokens $T_{\text{LiLaVe}}$ from which we extract the hidden state to train and evaluate the LiLaVe verifier:

$$L_{\text{LiLaVe}} = (-1, -2, -4, -8, -16),$$
$$T_{\text{LiLaVe}} = (-1, -2, -3, \ldots, -16).$$

As described in Section 2.1, in the LiLaVe's inference mode, for one LLM's decoding, we aggregate the XGBoost-inferred scores of hidden states corresponding to these tokens and layers using the arithmetic mean.

**Sampling temperature** When generating samples for the LiLaVe training, it is not immediately clear which sampling temperatures should be used. On one hand, for reasoning-intensive problems, low temperatures typically result in better performance. On the other hand, meta-generation techniques like majority voting require non-zero temperature to make the samples diverse (see Figure 13 in Appendix B demonstrating this trade-off for various datasets). Therefore, ideally, we want LiLaVe to perform well on samples generated across a range of temperatures. To check if it does, we experimentally study how the temperature of generations on which the verifiers are trained impacts their predictive ability when tested on answers to test questions, generated with various temperatures.

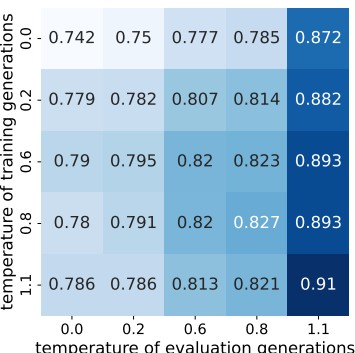

Fig. 2 shows the results of this analysis for hidden states of the Llama 3.1 8B and temperatures $\{0.0, 0.2, 0.6, 0.8, 1.1\}$ on GSM8K. Each cell represents a mean of 16 experiments: final AUC on the test set of 16 classifiers trained on hidden states from layers $\{-2, -4, -8, -16\}$ and tokens $\{-2, -4, -8, -16\}$. For all temperatures except 0, we generate 8 answers per question.

Figure 2: Performance (AUC) of LiLaVe trained and evaluated on hidden states of Llama 3.1 8B answers to GSM8K questions with various temperature settings.

---

[2] The area under the ROC curve (AUC) represents the probability that the model, if given a randomly chosen positive and negative example, will give a higher score to the positive example than to the negative one. Therefore, AUC is directly related to the downstream task performance of score-based methods like best-of-$n$ and weighted majority voting. For the conditional voting and self-correction methods, the score threshold for binary separation into positive / negative classes is additionally required, which needs to be tuned separately.

Table 1: Performance (AUC) of three methods for predicting the correctness of the LLM's answers: LiLaVe and four baseline methods: self-reflection, logprob-based confidence estimation, and two compute-intensive LLM-based ORMs finetuned either on Mistral-7B or DeepSeekMath-Instruct data.

| benchmark | LiLaVe | self-reflect | logprobs | ORM-Mistral | ORM-Deepseek |
|---|---|---|---|---|---|
| GSM8K (test) | 0.86 | 0.68 | 0.78 | 0.81 | **0.88** |
| GSM-Symbolic | 0.84 | 0.70 | 0.78 | 0.85 | **0.90** |
| GSM-Symbolic-p2 | **0.78** | 0.60 | 0.63 | 0.73 | 0.75 |
| algebra_linear_1d | **0.93** | 0.61 | 0.81 | 0.90 | 0.90 |
| MATH500 | 0.88 | 0.79 | 0.67 | 0.79 | **0.90** |

We observe that the predictive performance of the verifier increases both with the temperature of the evaluation samples as well as training samples. We hypothesize that increased temperature results in more diverse training examples and also examples with different correctness labels for one question, which is good for training the verifier. Higher temperature on the evaluation side likely results in samples that are incorrect in a way easier to detect by the verifier.

The experiment shows that increased temperature for generating training samples is beneficial. Given this result, and to ensure diversity in the training samples, we decide to train LiLaVe on samples generated with a mixture of five temperatures: $\{0, 0.25, 0.5, 0.75, 1.0\}$.

## 3.3 BASELINES

We compare LiLaVe with four baselines: two natural methods for estimating the correctness of the language model's answer – *logprob-based estimator* and *self-reflection prompting* – as well as two LLM-based verifiers. These baselines are described below, and their performance compared to LiLaVe is shown in Table 1.

**Logprob-based estimator** Assume that for a question $q$, a language model generates a response $y = y_1 y_2 \cdots y_n$, where each decoded token $y_i$ is given probability $p_i$. For each response, we compute the sum of log-probabilities over a $k$-suffix: $\sum_{i=0}^{k-1} \log p_{n-k}$.

We treat this sum as an (uncalibrated) estimator of the output correctness. The straightforward intuition behind it is that higher probabilities of the individual (suffix) tokens mean a higher probability of the answer. For each dataset, we choose the suffix length $k$, for which this estimator achieves the highest AUC score. We report results in Table 1. See Appendix B.4 for more details about this baseline, including a breakdown of performance over different suffix lengths.

**Self-reflection prompting** This baseline involves a base LLM self-reporting the confidence score (Tian et al., 2023; Pawitan & Holmes, 2024). Here we prompt the LLM (the same as the base one) to express a confidence of its answer being correct on a scale from 1 to 10. The specific self-reflection prompt we use is provided in Figure 16 in Appendix C.

**LLM-based verifiers** We also benchmarked two LLM-based verifiers (aka outcome reward models, or ORMs) developed in Xiong et al. (2024). Both of them are based on Llama 3.1 8B; they differ by the model that was used to generate their training data: either Mistral-7B or DeepSeekMath-Instruct 7B. The training datasets of both these ORMs consist of more than 250k. Note that LiLaVe was trained on *only 5k examples* per benchmark (5 samples for each of the 1k training questions).

Given the results in Table 1 comparing LiLaVe with the baselines, we conclude that LiLaVe excels at extracting useful signal, estimating the model's correctness of its answers. LiLaVe is significantly better than self-reflection prompting and logprob-based estimator. Moreover, LiLaVe achieves comparably good results as large, LLM-based verifiers trained on datasets a couple of orders of magnitude larger. Additionally, in Table 2 in Appendix B we present LiLaVe's strong results for two other base LLMs: Gemma 2 2B and Phi-3.5-mini.

## 3.4 LILAVE-BASED META-GENERATION STRATEGIES

As shown above, LiLaVe proves to be effective in distinguishing correct and incorrect LLM's responses as measured by the AUC metric. In this subsection, we experimentally demonstrate that this statistical performance can be translated into efficient and practical meta-generation strategies.

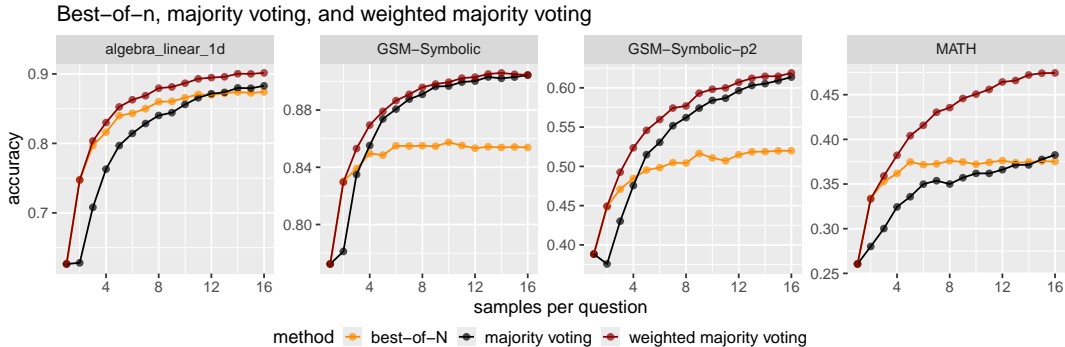

Figure 3: Best-of-$n$, majority voting, and weighted majority voting on four datasets. For each of the methods, the number of samples per question is varied between 1 and 16. Weighted majority voting performs best for all the datasets, but the margin differs across the datasets.

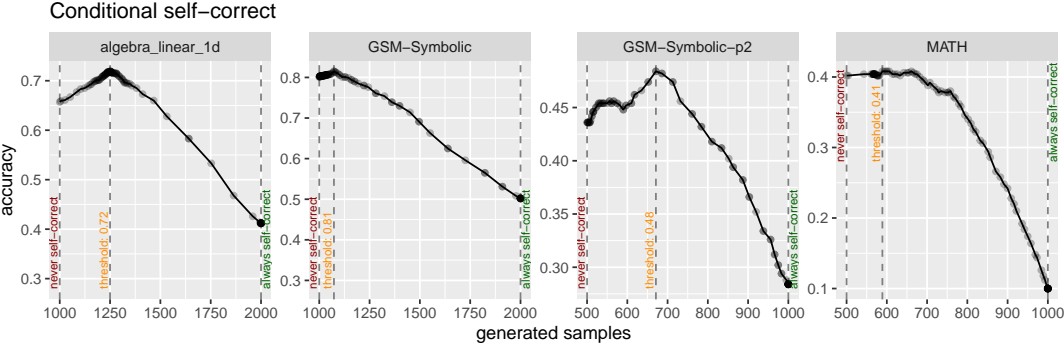

Figure 4: Conditional self-correction on four datasets. The dark points indicate the performance for different score thresholds. The left-most points correspond to no self-correction (and only one sample per question generated); The right-most points correspond to unconditional self-correction (and therefore two samples per question generated). The optimal thresholds are orange.

In this section, we show experiments using all the datasets from Section 3.1, except GSM8K – this is because, besides the previously mentioned weaknesses of this benchmark, Llama 3.1 8B achieves there results that are similar to the base version of GSM-Symbolic.

**Best-of-$n$ and weighted majority voting** First, we employ LiLaVe as a scoring function in best-of-$n$ and weighted majority voting strategies (see Section 2.2). For both strategies, we generate between 1 and 16 samples per question with temperature $1.0$, and score each of them with LiLaVe. In Figure 3, we show the results for both strategies, comparing them with the baseline of standard majority voting.

The weighted majority voting strategy performs best across all numbers of votes, and for all datasets, whereas for MATH, this dominance is the largest. For both GSM-Symbolic datasets, weighted majority voting is only slightly better than standard majority voting, and the difference diminishes with growing numbers of votes (samples). Best-of-$n$ is weaker than weighted majority voting, and for higher numbers of samples, also weaker than standard majority voting. This may be caused by *false positives*: responses appearing as correct to the verifier; the chance of encountering such examples grows with the number of samples (*cf* Section 5.1 of (Cobbe et al., 2021)).

**Conditional self-correction** We evaluate the conditional self-correction strategy (Section 2.2) with a sampling temperature of 0. Figure 4 shows the performance across four datasets for varying thresholds $s \in [0, 1]$, which control how often self-correction is attempted.

A typical issue with self-correction is that while LLMs are often able to fix incorrect responses, they also turn many correct responses into incorrect ones. As seen in Figure 4, applying self-correction to all responses reduces accuracy by 15–30 percentage points. However, selectively correcting only low-scoring responses leads to significant gains for algebra_linear_1d and GSM-Symbolic-p2, with

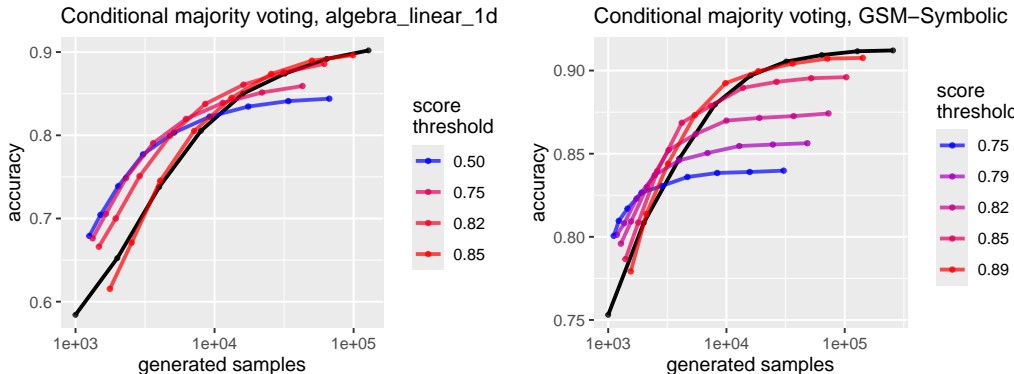

Figure 5: Conditional majority voting with varying threshold $s$ and the number of samples per question $n$ between 1 and 256. Different colors refer to different thresholds $s$. The parameter $n$ is shown implicitly as for fixed $s$ it influences the total number of generated samples through the number of dataset questions scored below $s$ (which for dataset $\mathcal{D}$ is equal $(n+1) \cdot |\mathcal{D}|$; the additional one sample per example is the probe sample). Note that on the $x$-axis, a logarithmic scale is applied. In black, the baseline of standard majority voting is shown. Conditional majority voting outperforms the baseline on a wide range of generation budgets.

smaller improvements on other datasets. The optimal threshold varies per dataset (indicated in orange in Figure 4), so in practice, this hyperparameter must be tuned depending on the data.

**Conditional majority voting** In this meta-generation strategy (Section 2.2) four hyperparameters are involved: the temperature $t_0$ of generating the probe sample, the temperature of the samples for majority voting $t_{mv}$, the score threshold $s$ below which the majority voting is triggered, and the number of majority voting samples $n$. We fix $t_0 = 0, t_{mv} = 1$, and perform experiments with $n \in \{1, 2, 4, \ldots, 256\}$ and a range of various $s \in [0, 1]$. Figure 5 presents results for two datasets.

In these plots, we do not explicitly show the $n$ parameter, but instead, on the $x$ axis, we put the total number of samples generated when evaluating on all the examples (which is influenced by both $n$ and $s$). This exposes an interesting fact: for a fixed budget (in terms of the number of generated samples), different combinations of $n$ and $s$ parameters of conditional majority voting give optimal accuracy. Importantly, conditional majority voting for lower budgets achieves better performance than standard majority voting (black line in the plots). This shows that LiLaVe-conditioned majority voting is a practical method allowing for trade between accuracy and efficiency in restricted budget settings. In a real scenario, one would tune the $n$ and $s$ parameters on a validation set to achieve a desired accuracy-efficiency trade-off.

## 4 RELATED WORK

**Reasoning and large language models** Step-by-step problem solving is fundamental to human intelligence and scientific discovery. Mathematical problems are often considered a hallmark of reasoning and have been extensively studied in the context of LLMs (Lewkowycz et al., 2022; Cobbe et al., 2021; Hendrycks et al., 2021). The field is advancing rapidly, with models like OpenAI's o3 solving certain research-level problems from the FrontierMath benchmark (Glazer et al., 2024). Although o3's training details remain undisclosed, conjecturally similar DeepSeek-R1 (Guo et al., 2025) exemplifies the class of "thinking models," typically trained with reinforcement learning to conduct extensive searches over the space of solutions. The flip side is the high inference cost; o3 reportedly used 33M tokens to solve a single ARC-AGI puzzle (Chollet, 2019; Chollet et al., 2024). This underscores the need for efficient inference, which has become a growing research focus. Snell et al. (2024) and Wu et al. (2025) explore trade-offs between model size and inference time, aiming to establish compute-optimal strategies. Our work similarly prioritizes inference-time efficiency, with a particular focus on reward model design.

**Inference-time techniques** Chain-of-Thought (CoT) prompting (Wei et al., 2022; Nye et al., 2021) is arguably the most widely adopted technique for improving LLM reasoning. Self-consistency

decoding (Wang et al., 2023) involves generating multiple answers and applying majority voting. Furthermore, tree and graph search methods, including Monte Carlo tree search and AlphaZero-inspired techniques, have been widely studied (Yao et al., 2023; Besta et al., 2024; Feng et al., 2024; Welleck et al., 2022). Another research direction focuses on self-refinement techniques where the LLM responses are iteratively improved / fixed by the model itself, possibly using external feedback (Havrilla et al., 2024; Madaan et al., 2023; Shinn et al., 2023). However, the effectiveness and efficiency of these methods remain limited (Huang et al., 2024; Havrilla et al., 2024). Our work contributes to the area of inference-time techniques by proposing a lightweight verifier that can boost the accuracy of the base language model with low computational overhead. For a broad overview of inference-time generation techniques with large language models, see (Welleck et al., 2024).

**Approximate verifiers** LLM-generated answers or reasoning process can be assessed by fine-tuned models, known either as *verifiers* or *reward models*. Verifiers can be trained to predict correctness of entire answers (Cobbe et al., 2021) or to verify individual reasoning steps (Lightman et al., 2023; Yu et al., 2024; Havrilla et al., 2024; Uesato et al., 2022).[3] Acquiring training data remains the key challenge. Lightman et al. (2023) rely on costly human data, while Wang et al. (2024a), Wang et al. (2024b), Luo et al. (2024), and Havrilla et al. (2024) generate synthetic data. A recent work Ye et al. (2024) examines LLM reasoning rationales and hidden mechanisms, suggesting that latent structures could enable training simple verifiers, which inspired our work.

**Probing** Probing (Alain & Bengio, 2018) the internal states of transformer models has become an established method of studying their latent representations (Gurnee & Tegmark, 2024), memorized sensitive information (Kim et al., 2023), and in-context algorithms (Akyürek et al., 2023). For a recent introduction to techniques for studying the internal workings of transformer-based language models, see (Ferrando et al., 2024). Using the models' hidden layer activations to predict the truthfulness of their generations has been extensively studied in the context of hallucination detection, see (Azaria & Mitchell, 2023; Chen et al., 2024; He et al., 2024; Beigi et al., 2024). Outside of hallucination detection, OPENIA (Bui et al., 2025) notes that model internal representations encode information useful for predicting the correctness of generated code. While applying this insight to a different domain, we also use a different type of latent classifier and additionally study recipes for utilizing the verifiers to improve model generations.

## 5 LIMITATIONS AND FUTURE WORK

**Verifier-conditioned decoding** In our experiments, the LiLaVe verifier scores answers after full generation. However, as shown in Figure 1, LiLaVe detects useful signal throughout the sequence, even at the first decoded token. This suggests integrating the verifier directly into decoding as a *reward model* guiding token selection toward high-certainty paths while avoiding erroneous trajectories. Given LiLaVe's efficiency and low computational overhead, this direction is particularly promising.

**Verifier-oracle gap** While our work advances test-time reasoning, there is still substantial room for improvement. In the best-of-$n$ setting, when an oracle selects a correct answer if present among $n$ samples, performance increases dramatically (see Figure 10 in Appendix B). This performance gap highlights the potential for improving verifiers, which could translate to significant gains.

We hypothesize that better verifiers can be obtained by integrating information for a larger number of tokens and layers, as well as creating ensemble models utilizing additional information such as logits and self-evaluation, which we treat as separate methods here.

Moreover, LiLaVe can be combined with different base LLMs which may constitute multiple *standalone verifiers* that digest responses generated beforehand from an arbitrary model. See Appendix B.7 for a prototype experiment in that direction, which gave promising results.

**Adaptive conditional majority voting** In our conditional majority voting strategy, we fix the number of samples $n$ to be generated per question beforehand. This could be optimized by allowing $n$ to be selected *adaptively*, based on the score from the verifier. Our initial experiments have shown promising results: the verifier's score on the probe sample was inversely correlated with the entropy among the answers in the subsequently generated samples. This suggests a meta-generation strategy where lower scores of the probe sample imply larger numbers of samples for voting.

---

[3]The verifiers for entire answers are also called *outcome reward models* (ORM), whereas the verifiers for reasoning steps are called *process reward models* (PRM).

## REPRODUCIBILITY STATEMENT

In the Supplementary material we provide code and data for reproducing LiLaVe experimental results presented in this paper. In particular, we include:

- The four datasets used in the experiments (GSM8K, GSM-Symbolic, MATH, algebra_linear_1d) including the training / testing splits.
- A script for extracting hidden states from the base LLMs to train LiLaVe models.
- A script for training LiLaVe's XGBoost model based on the hidden states.
- Three pre-trained LiLaVe models: `algebra_linear_1d.xgb`, `GSM8K.xgb`, and `MATH.xgb`, trained on the training partitions of the respective datasets.
- The implementation of the meta-generation strategies described in the paper.
- Prompts we used for LLM generations.

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

> **Question:** Travis has 10000 apples, and he is planning to sell these apples in boxes. Fifty apples can fit in each box. If he sells each box of apples for $35, how much will he be able to take home?
>
> - - - - - - - - - - - - - - - - - - - - - - - - - - - - - - - - - - - - - - - - - - - - - -
>
> **Rationale:** The total of boxes of apples is 10000 / 50 = 200. Therefore the total amount he can take home is $200 \times \$35 = 7000$.
> **Answer:** 7000

Figure 6: An example of a question from the GSM8K benchmark, followed by a couple of reasoning steps – a rationale for the final answer, which is always a number.

## A  REASONING-FOCUSED BENCHMARKS

We evaluate LiLaVe and LiLaVe-based meta-generation strategies on four mathematical QA datasets. For each of them we select 1000 training examples to train a dataset-specific LiLaVe. We test on sets of 500–1319 examples, depending on the dataset. Below, we describe each of the benchmarks. We share the data partitions in the supplementary materials.

**GSM8K** Cobbe et al. (2021), contains grade school math problems with integer answers. To fit LiLaVe, we select 1000 examples from its training partition, and in evaluation, use its full test set of 1319 questions. For answer generation, we use the standard 8-shot chain-of-thought prompt used in Wei et al. (2022). While widely used in LLM reasoning research, GSM8K is a relatively easy benchmark for modern LLMs. Also, it is likely leaked into LLM pretraining data.

**GSM-Symbolic** Mirzadeh et al. (2024), has been developed to mitigate data contamination problem of GSM8K by semi-automatically generating questions from question templates obatined from GSM8K. Additional variants **p1** and **p2** of this dataset add one or two extra clauses to questions, increasing reasoning complexity. When evaluating LiLaVe-based generation strategies on GSM-Symbolic, we reuse GSM8K's training set for training the verifier. We also apply the same 8-shot chain-of-thought prompt that we use for GSM8K.

**algebra_linear_1d** is a subset of a synthetic benchmark introduced in Saxton et al. (2019) to evaluate the performance of language models on a broad range of common mathematical tasks. algebra_linear_1d evaluates models for solving single-variable linear equations with integer solutions. We generate training and test sets, each containing 1000 examples. To query an LLM for answers, we use a simple zero-shot CoT prompt (see Figure 17). In Figure 7 (in Appendix B), there is an example of a question and solution from algebra_linear_1d.

**MATH** Hendrycks et al. (2021) contains competition-level mathematical problems. We train LiLaVe on 1000 selected training questions, and in evaluation we use its **MATH500** subset used in Lightman et al. (2023). LLM inference is performed using the 4-shot chain-of-thought prompt used in Lewkowycz et al. (2022). The final answers to MATH's questions include expressions such as polynomials, fractions, or complex numbers. To evaluate the generated answers, they need to be properly parsed and semantically compared with the ground truth. For that, we reuse the final answer extractor from Gao et al. (2024).[4]

## B  ADDITIONAL RESULTS

### B.1  TEMPERATURE OF GENERATIONS

Our latent verifiers are trained on hidden states gathered from intermediate layers of the LLama 3.1 8B model, using generations sampled at different temperatures. In this section, we extend the temperature sensitivity analysis (discussed earlier in Section 3.2) beyond GSM8K to additional datasets. We test how well do verifiers trained on various temperatures transfer to verifying generations sampled from

---

[4]Specifically, we reuse the code available at `https://github.com/EleutherAI/lm-evaluation-harness/blob/main/lm_eval/tasks/minerva_math/utils.py`

**Question:** Solve $-78 = 30 * r + 150 - 78$ for $r$.

---

**Rationale:** First, let's simplify the right-hand side of the equation by combining the constants:

$$-78 = 30 * r + 72$$

Next, let's subtract 72 from both sides of the equation to isolate the term with $r$:

$$-78 - 72 = 30 * r$$
$$-150 = 30 * r$$

Now, let's divide both sides of the equation by 30 to solve for r:

$$-150/30 = r$$
$$-5 = r$$

**Answer:** -5

Figure 7: An example of a question from the algebra_linear_1d benchmark, and a solution followed by a correct answer generated by Llama 3.1 8B.

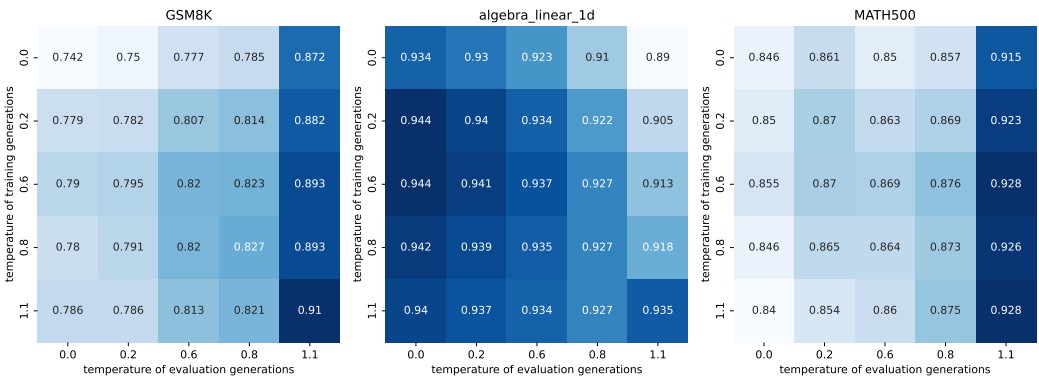

Figure 8: Transfer of performance (AUC) of LiLaVe trained and evaluated on hidden states extracted from answers generated from different temperatures, for three datasets. The base LLM used in this experiment is Llama 3.1 8B.

different ones. The heatmap for GSM8K is identical to the one presented in Figure 5, we include it here again for the comparizon with results on other datasets.

Figure 8 shows heatmaps with the results of this analysis for hidden states of LLama 3.1 8B model and temperatures from the set $0.0, 0.2, 0.6, 0.8, 1.1$ on three datasets: GSM8K, algebra_linear_1d, and MATH500. For all temperatures, except temperature 0, we generate 8 answers for each question. Each cell in a heatmap represents the mean AUC of XGBoost verifier on an appropriate test set, averaged over 16 classifiers trained on hidden states from different layers $(2, 4, 8, 16)$ and different tokens $(2, 4, 8, 16$, counted from the end of the generated sequence).

Observations and conclusions for GSM8K are discussed in Section 3.2. Most importantly, the predictive performance of the verifier increases with both the training and evaluation temperatures. For algebra_linear_1d, most AUC values are very close to each other, but the variability trend differs from GSM8K: for a fixed training temperature, the verifier performs better when the evaluation temperature is lower. The heatmap for MATH follows a similar pattern to GSM8K, but the optimal training temperature for a fixed evaluation temperature is reached faster – AUC increases until $T = 0.6$ and then plateaus.

## B.2 CHOICE OF LiLaVe ARCHITECTURE

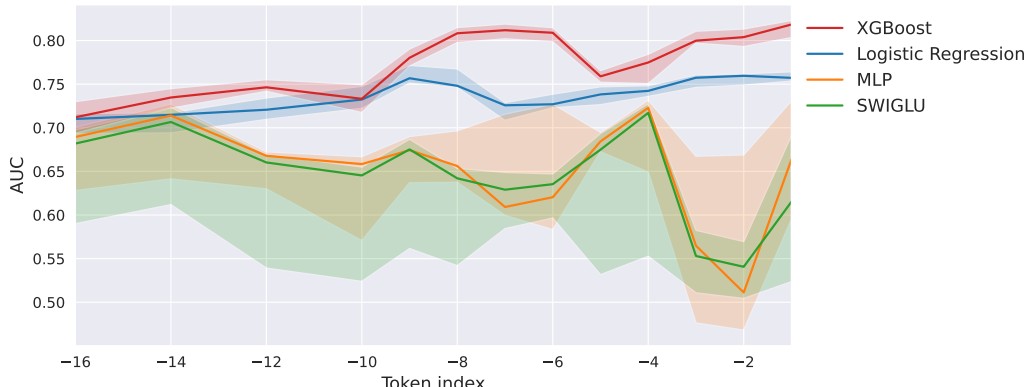

Figure 9: Ablation on the architecture of LiLaVe. Methods are compared on generations from Llama 3.1 8B on GSM8K dataset. The x-axis represents token index, while y-axis represents the value of AUC metric.

In all our main experiments, we instantiate our verifier as an XGBoost model Chen & Guestrin (2016). This choice is informed by our ablation experiments, which demonstrate its superior performance compared to alternative architectures. Additionally, XGBoost requires minimal hyperparameter tuning, making it a practical choice. We set the maximum tree depth to 5, selecting it as one of several equally well-performing candidates, and we use a learning rate (eta) of 0.1. All other hyperparameters are the default ones. Training a single instance of XGBoost classifier in our setup is computationally efficient, taking only three minutes on our CPUs.

To validate our choice, we compare XGBoost against three other methods: Logistic Regression, a Multi-Layer Perceptron (MLP), and a SWIGLU-based MLP Shazeer (2020). Each method is trained on token-level features extracted from the token $T$ ($T \in \{-1, -2, \ldots, -16\}$) and the layer $L$ ($L \in \{-1, -2, \ldots, -5\}$). We run each method for each $T$ and $L$ for 10 seeds. Figure 9 illustrates the comparative performance of these architectures, highlighting XGBoost's consistent superiority over the alternatives. For each line and plot The solid lines are medians, and the shadow region is a nonsymmetric 90% confidence interval.

While hyperparameter tuning for MLP and SWIGLU could potentially improve their performance, we performed only a limited sweep over the number of layers and learning rates. However, the difficulty of tuning these models further underscores the advantage of XGBoost, which performs well out-of-the-box with minimal effort.

### B.2.1 HYPERPARAMETERS OF COMPARED METHODS

**Logistic Regression** We use an `sklearn` implementation with a maximum iteration count of 1000 and balanced classes.

**MLP** The MLP consists of a hidden layer of size 16, and an output dimension of 1. It is trained using a logistic regression loss for 20 epochs with a batch size of 32. The model is optimized with Adam, using a learning rate of $10^{-4}$.

**SWIGLU** This variant is a residual MLP using SWIGLU activations. It has two hidden layers of size 32, and an intermediate hidden dimension of 16. Like the standard MLP, it is trained with logistic regression loss for 20 epochs and a batch size of 32. The learning rate is $5 \times 10^{-4}$, and weight decay is set to 0.1.

**XGBoost** In all our experiments, we train XGBoost with the following hyperparameters:

- max_depth=10,
- eta=0.1,

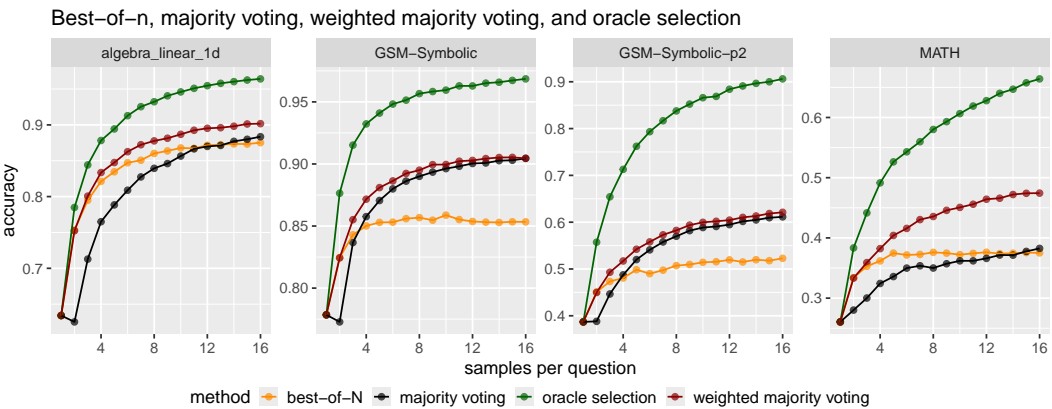

Figure 10: Comparison of meta-generation strategies to oracle selection.

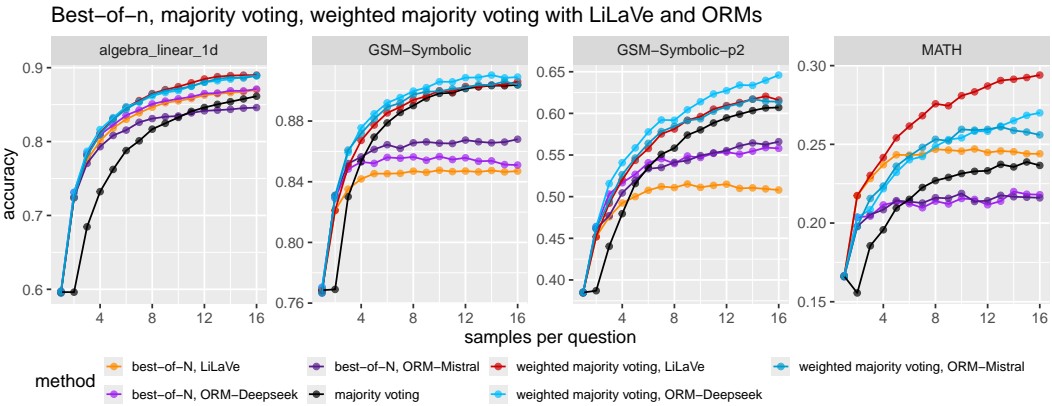

Figure 11: Comparison of meta-generation strategies with LiLaVe and two LLM-based ORMs.

- nrounds=30.

The rest of the hyperparameters use their default values set by the authors of the official XGBoost implementation.

All input sizes are equal to 4096, as this is the dimensionality of Llama 3.1 8B hidden states. For MLP and SWIGLU, we report their test performance on an epoch after which the validation performance is the best.

### B.3 ACCURACY OF BEST-OF-N GIVEN THE ORACLE

We compare the results of meta-generation strategies to oracle selection. This theoretical and practically impossible strategy assumes access to an omnipotent verifier, which always selects the correct answer from the set of LLM-generated ones, if only such a correct answer appears in this set. Otherwise, the strategy fails. Figure 10 presents the results of this experiment, suggesting a gap between the best known meta-generation strategy and this theoretical upper bound, suggesting that further improving the verifiers still has a lot of potential.

### B.4 LOGPROBS BASELINE RESULTS

This section provides a detailed analysis of the logprob-based estimator introduced in Section 3.3. The estimator is computed as the sum of log probabilities from Llama 3.1 8B over the final $k$ tokens of a generated answer.

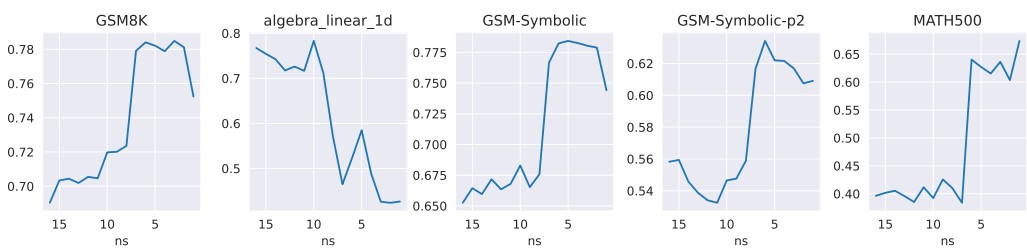

Figure 12: AUC of the sum of log probabilities over the answer suffix. The results correspond to zero-temperature generations from LLaMA 3.1 8B on the test sets of the respective datasets. The x-axis represents the length of the suffix considered.

Figure 13: Accuracy of majority voting for different generation temperatures and number of votes. Base model is Llama 3.1 8B.

Figure 12 shows the AUC scores of this estimator across various (1-16) suffix lengths and datasets. For each dataset, we select the suffix length $k$ that yields the highest AUC, and report these results in Table 1. Thus, this table reflects the best-performing suffix length for each dataset, giving an idealized upper bound on the estimator's performance.

In most datasets (GSM8K, GSM-Symbolic, GSM-Symbolic-p2, and MATH500), we observe a positive correlation between model confidence (measured by the sum of log probabilities) and answer correctness: a higher confidence in the final tokens generally indicates a higher chance of correctness. Interestingly, an exception arises in the algebra_linear_1d dataset, where the relationship is inverted. Specifically, for short suffixes (lengths 1 to 8), AUC falls below 0.5. This implies that in this dataset, higher model confidence is actually indicative of a greater chance of error, suggesting the base model is overconfident.

Since the suffix length $k$ is fixed, normalization of the sum is not necessary. We also verified that this sum-based estimator consistently outperforms a more commonly used average over all logprobs in the answer.

### B.5 OPTIMAL TEMPERATURES IN MAJORITY VOTING

In this experiment, we evaluate the accuracy of majority voting (see Section 3.4) with respect to the temperature of generations and the number of votes. Results are presented in Figure 13. We observe that for different numbers of votes, different generation temperatures are optimal.

### B.6 PERFORMANCE OF LILAVE WITH OTHER BASE LLMS

In Table 2 we present AUC performance of LiLaVe for three different base LLM: Llama 3.1 8B (used in the main experimental line presented in the main text), Gemma 2 2B, and Phi-3.5-mini.

Table 2: Performance (AUC) of LiLaVe for three different base LLMs: Llama 3.1 8B, Gemma 2 2B, Phi-3.5-mini. LiLaVe preserves strong predictive performance across all the three models and all the benchmarks – with one exception of Gemma on MATH. The reason is likely because this model scored only ∼5% on MATH, which did not give enough positive examples for training LiLaVe.

| benchmark | Llama 3.1 8B | Gemma 2 2B | Phi-3.5-mini |
|---|---|---|---|
| GSM8K (test) | 0.86 | 0.83 | 0.83 |
| GSM-Symbolic | 0.84 | 0.83 | 0.79 |
| GSM-Symbolic-p2 | 0.78 | 0.84 | 0.78 |
| algebra_linear_1d | 0.93 | 0.86 | 0.96 |
| MATH500 | 0.88 | 0.53 | 0.93 |

Table 3: A comparison of two verification setups: the standard one, where responses generated by Phi-3.5-mini are scored based on the hidden states extracted from Phi during the generation (the middle column), *versus* responses generated by Phi, but later ingested by LLama 3.1 8B and scored based on the hidden states extracted from it (the right column). The latter setup in general performs worse – but not much worse, and for GSM-Symbolic actually better.

| benchmark | Phi-3.5-mini + Phi-LiLaVe | Phi-3.5-mini + Llama-LiLaVe |
|---|---|---|
| GSM8K (test) | **0.83** | **0.83** |
| GSM-Symbolic | 0.79 | **0.83** |
| GSM-Symbolic-p2 | **0.78** | 0.76 |
| algebra_linear_1d | **0.96** | 0.94 |
| MATH500 | **0.93** | 0.91 |

### B.7 PERFORMANCE OF LILAVE TESTED ON RESPONSES ORIGINATING FROM A DIFFERENT MODEL

The standard mode of using LiLaVe is to apply it to the hidden states of the base LLM that generates the response. However, another setup is possible, where the responses are given without the hidden states and these are recreated by digesting the responses by an LLM for which a LiLaVe is available. In Table 3 we compare the results of two such approaches. The responses are coming from Phi-3.5-mini, and they are scored either by the LiLaVe trained for Phi (Phi-LiLaVe), or by Llama-LiLaVe, after retrieving the hidden states from Llama 3.1 8B that digested the Phi's responses. As can be seen, the latter setup gives good results, only slightly weaker than the original setup.

### B.8 TRANSFER TO OTHER DATASETS

We evaluate the generalization ability of a verifier trained on one dataset when applied to another. Table **??** presents the AUC scores for different train-test combinations.

Our results indicate that while training and evaluating on the same dataset yields the highest performance, there is a significant cross-dataset generalization. For instance, a verifier trained on GSM8K achieves an AUC of 0.87 on algebra_linear_1d and 0.84 on MATH, which is better then baseline methods based on logprobs and self-reflection (see Table 1). Interestingly, the verifier trained on MATH generalizes less effectively, achieving only 0.53 AUC on algebra_linear_1d and 0.72 on GSM8K.

Overall, the results of this experiment suggest that some transferability across datasets exists, but we leave the exploration of transferability to other models for future work.

### B.9 ADDITIONAL DATASETS AND MODELS (QWEN 3 AND AIME)

We run an experiment where we use a new, strong reasoning model, Qwen3 8B, as the base LLM, and more challenging math dataset: AIME. We train LiLaVe on Qwen3 responses to problems from AIME 1983-2021 and test on AIME 2022-2025. We also run Qwen3 both in "thinking" (long-CoT)

Table 4: Transfer of performance (AUC) of LiLaVe trained and evaluated on different datasets. The base LLM used in this experiment is Llama 3.1 8B.

| test / train | algebra_linear_1d | GSM8K | MATH |
|---|---|---|---|
| algebra_linear_1d | **0.93** | 0.75 | 0.71 |
| GSM8K | 0.87 | **0.86** | 0.84 |
| MATH | 0.53 | 0.72 | **0.88** |
| algebra_linear_1d + GSM8K | **0.94** | **0.85** | 0.84 |
| GSM8K + MATH | 0.81 | **0.85** | **0.88** |
| MATH + algebra_linear_1d | **0.93** | 0.81 | **0.88** |
| algebra_linear_1d + GSM8K + MATH | **0.93** | 0.84 | 0.87 |

Table 5: AUC performance of LiLaVe with Qwen and Llama models on AIME and GPQA-Diamond datasets.

| model | AIME | GPQA-Diamond |
|---|---|---|
| Llama 3.1 8B | 0.67 | 0.76 |
| Qwen3 thinking | 0.97 | – |
| Qwen3 non-thinking | 0.96 | – |

and "non-thinking" (standard-CoT) mode. We show the results in Table 5. LiLaVe achieved excellent AUC verification performance for both modes:

When we use Llama 3.1 8B as the base LLM in the same setting, LiLaVe achieves a much weaker AUC of 0.67. The main reason likely is the fact that Llama gets only 12% accuracy on AIME which results in too few positives for training LiLaVe. Qwen3 gets 34% and 65% of accuracy in non-thinking and thinking mode, respectively. Moreover, Qwen3 is a strong reasoning model on its own and when it cannot produce a correct answer, it may manifest quite clearly in the hidden states.

## B.10 COMPARISON WITH A PRM

We ran additional experiments comparing the performance of LiLaVe to Math-Shepherd. We used the exact same data (Llama 3.1 8B math solutions) as for the experiment whose results we present in Table 1. Below, we compare the performance of LiLaVe and Math-Shepherd. We also include the performance of the stronger ORM from Table 1. We present the results in Table 6.

The results are mixed: Math-Shepherd performs better on GSM-style tasks, while LiLaVe does better on MATH500 and linear algebra. Also, in accordance with the literature, Math-Shepherd performs better than the ORM.

In general, we find it interesting and positive that even heavy-weight PRMs sometimes are weaker than lightweight LiLaVe operating on hidden states and trained on relatively small training sets.

Table 6: LiLaVe vs Math-Shepherd PRM.

| benchmark | LiLaVe | ORM-Deepseek | Math-Shepherd |
|---|---|---|---|
| GSM8K (test) | 0.86 | 0.88 | 0.89 |
| GSM-Symbolic | 0.84 | 0.90 | 0.91 |
| GSM-Symbolic-p2 | 0.78 | 0.75 | 0.79 |
| algebra_linear_1d | 0.93 | 0.90 | 0.92 |
| MATH500 | 0.88 | 0.90 | 0.82 |

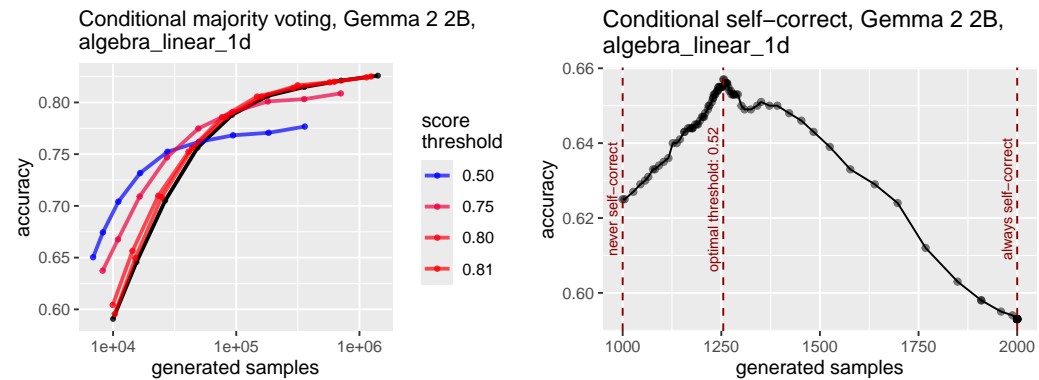

Figure 14: Conditional majority voting and conditional self-correction for **Gemma 2 2B** model, on **algebra_linear_1d** benchmark. The results are similarly good as those for Llama 3.1 8B.

> The solution you provided contains mistakes and the answer is incorrect. Please, carefully review the solution and write a new, correct one.

Figure 15: Prompt used for the self-correction experiments.

## C   PROMPTS

We present prompts used in our experiments in Figure 15, Figure 16, Figure 18, Figure 17, and Figure 19.

## D   EFFICIENCY

LLM-based verifiers typically require a large number of training examples, e.g. both models from (Xiong et al., 2024) which we benchmark against, were trained on over 250k examples. In contrast, LiLaVe achieves comparable performance with just 5k samples per benchmark – two orders of magnitude less – making it a strong choice in data-scarce settings. Once the hidden states are collected, training LiLaVe takes only 15 minutes on a CPU, compared to the GPU-intensive fine-tuning required for LLM-based verifiers.

In terms of inference efficiency, scoring pre-generated Llama 3.1 8B's responses to 1319 GSM8K test questions using an LLM-based verifier (via the code from (Xiong et al., 2024)) took nearly 20 minutes on an NVIDIA GH200 GPU. The same task (having the hidden states extracted) was completed by LiLaVe in only $\sim 3.4$s of wall clock time on CPUs of a Dell Precision 3561 laptop, yielding a $\sim 350\times$ speedup.

Of course, one could argue that, like other verifiers, LiLaVe still relies on a large generator to produce the answer to be verified. In scenarios where both generation and verification are benchmarked together, the speedup offered by LiLaVe may be limited to at most $2\times$, assuming verifier and generator are of similar size. However, even in this setting, LiLaVe provides important practical advantages. Unlike LLM-based verifiers that require GPUs, LiLaVe runs efficiently on CPU. This avoids the need to load large generator and verifier onto separate GPUs, which would double the required hardware, or to repeatedly load and unload model weights to and from GPU memory, which can significantly

> Please, rate on a scale of 1 to 10 how confident you are of the correctness of your answer.

Figure 16: Prompt used for the self-reflection confidence estimation.

> Think step by step.

Figure 17: 0-shot prompt for the algebra_linear_1d dataset.

> Question: There are 15 trees in the grove. Grove workers will plant trees in the grove today. After they are done, there will be 21 trees. How many trees did the grove workers plant today?
> Answer: There are 15 trees originally. Then there were 21 trees after some more were planted. So there must have been 21 - 15 = 6. The answer is 6.
> ─────────────────────────────────────────────
> Question: If there are 3 cars in the parking lot and 2 more cars arrive, how many cars are in the parking lot?
> Answer: There are originally 3 cars. 2 more cars arrive. 3 + 2 = 5. The answer is 5.
> ─────────────────────────────────────────────
> Question: Leah had 32 chocolates and her sister had 42. If they ate 35, how many pieces do they have left in total?
> Answer: Originally, Leah had 32 chocolates. Her sister had 42. So in total they had 32 + 42 = 74. After eating 35, they had 74 - 35 = 39. The answer is 39.
> ─────────────────────────────────────────────
> Question: Jason had 20 lollipops. He gave Denny some lollipops. Now Jason has 12 lollipops. How many lollipops did Jason give to Denny?
> Answer: Jason started with 20 lollipops. Then he had 12 after giving some to Denny. So he gave Denny 20 - 12 = 8. The answer is 8.
> ─────────────────────────────────────────────
> Question: Shawn has five toys. For Christmas, he got two toys each from his mom and dad. How many toys does he have now?
> Answer: Shawn started with 5 toys. If he got 2 toys each from his mom and dad, then that is 4 more toys. 5 + 4 = 9. The answer is 9.
> ─────────────────────────────────────────────
> Question: There were nine computers in the server room. Five more computers were installed each day, from monday to thursday. How many computers are now in the server room?
> Answer: There were originally 9 computers. For each of 4 days, 5 more computers were added. So 5 * 4 = 20 computers were added. 9 + 20 is 29. The answer is 29.
> ─────────────────────────────────────────────
> Question: Michael had 58 golf balls. On tuesday, he lost 23 golf balls. On wednesday, he lost 2 more. How many golf balls did he have at the end of wednesday?
> Answer: Michael started with 58 golf balls. After losing 23 on tuesday, he had 58 - 23 = 35. After losing 2 more, he had 35 - 2 = 33 golf balls. The answer is 33.
> ─────────────────────────────────────────────
> Question: Olivia has $23. She bought five bagels for $3 each. How much money does she have left?
> Answer: Olivia had 23 dollars. 5 bagels for 3 dollars each will be 5 x 3 = 15 dollars. So she has 23 - 15 dollars left. 23 - 15 is 8. The answer is 8.
> ─────────────────────────────────────────────
> Question:

Figure 18: 8-shot prompt for GSM8K, GSM-Symbolic, and GSM-symbolic-p2 datasets.

Problem: Find the domain of the expression $\frac{\sqrt{x-2}}{\sqrt{5-x}}$.

Solution: The expressions inside each square root must be non-negative. Therefore, $x-2 \geq 0$, so $x \geq 2$, and $5 - x \geq 0$, so $x \leq 5$. Also, the denominator cannot be equal to zero, so $5 - x > 0$, which gives $x < 5$. Therefore, the domain of the expression is $\boxed{[2,5)}$. Final Answer: The final answer is $\boxed{[2,5)}$. I hope it is correct.

- - - - - - - - - - - - - - - - - - - - - - - - - - - - - - - - - - - - - - -

Problem: If $\det \mathbf{A} = 2$ and $\det \mathbf{B} = 12$, then find $\det(\mathbf{AB})$.

Solution: We have that $\det(\mathbf{AB}) = (\det \mathbf{A})(\det \mathbf{B}) = (2)(12) = \boxed{24}$. Final Answer: The final answer is $\boxed{24}$. I hope it is correct.

- - - - - - - - - - - - - - - - - - - - - - - - - - - - - - - - - - - - - - -

Problem: Terrell usually lifts two 20-pound weights 12 times. If he uses two 15-pound weights instead, how many times must Terrell lift them in order to lift the same total weight?

Solution: If Terrell lifts two 20-pound weights 12 times, he lifts a total of $2 \cdot 12 \cdot 20 = 480$ pounds of weight. If he lifts two 15-pound weights instead for $n$ times, he will lift a total of $2 \cdot 15 \cdot n = 30n$ pounds of weight. Equating this to 480 pounds, we can solve for $n$:

$$30n = 480$$

$$\Rightarrow \qquad n = 480/30 = \boxed{16}$$

Final Answer: The final answer is $\boxed{16}$. I hope it is correct.

- - - - - - - - - - - - - - - - - - - - - - - - - - - - - - - - - - - - - - -

Problem: If the system of equations

$$6x - 4y = a,$$
$$6y - 9x = b.$$

has a solution $(x, y)$ where $x$ and $y$ are both nonzero, find $\frac{a}{b}$, assuming $b$ is nonzero.

Solution: If we multiply the first equation by $-\frac{3}{2}$, we obtain

$$6y - 9x = -\frac{3}{2}a.$$

Since we also know that $6y - 9x = b$, we have

$$-\frac{3}{2}a = b \Rightarrow \frac{a}{b} = \boxed{-\frac{2}{3}}.$$

Final Answer: The final answer is $\boxed{-\frac{2}{3}}$. I hope it is correct.

- - - - - - - - - - - - - - - - - - - - - - - - - - - - - - - - - - - - - - -

Problem:

Figure 19: 4-shot prompt for the MATH dataset.

slow down the whole pipeline. It also introduces minimal compute overhead compared to LLM-based verifiers, which makes it much easier to integrate with more adaptive generation strategies.

