# OpenReview forum: "Lightweight Latent Verifiers for Efficient Meta-Generation Strategies"
_ICLR.cc/2026/Conference — Submitted to ICLR 2026_

### Official Review · Reviewer_KwTJ · 2025-10-16

**Soundness:** 3
**Presentation:** 3
**Contribution:** 3
**Rating:** 6
**Confidence:** 3

**Summary:**

This paper introduces LiLaVe (Lightweight Latent Verifier), a novel approach for verifying the correctness of LLM-generated solutions to reasoning-intensive tasks by extracting signals from the hidden states of the base language model. The key innovation is training a lightweight XGBoost classifier on hidden states extracted from specific layers and token positions during generation, labeled by answer correctness. The authors demonstrate that LiLaVe achieves competitive verification performance with large LLM-based verifiers while requiring only approximately 5k training examples and running efficiently on CPU. The paper further introduces practical meta-generation strategies leveraging LiLaVe, including conditional majority voting and conditional self-correction, which improve both accuracy and computational efficiency on mathematical reasoning benchmarks (GSM8K, GSM-Symbolic, MATH, algebra_linear_1d).
The work provides both theoretical insights into what information is encoded in LLM hidden states and practical contributions toward resource-efficient verification for reasoning tasks. The authors conduct extensive ablation studies to determine optimal extraction locations (layers and tokens) and demonstrate effectiveness across multiple base models (Llama 3.1 8B, Gemma 2 2B, Phi-3.5-mini).

**Strengths:**

- The paper addresses a practically important problem of expensive LLM-based verification with a simple and effective solution that achieves comparable performance while being orders of magnitude faster.
- The experimental evaluation is comprehensive, covering four different benchmarks, three base LLMs, and multiple meta-generation strategies with clear performance improvements demonstrated in Figures 3-5.
- The systematic analysis of which hidden states (layers and tokens) contain optimal correctness signals in Section 3.2 and Figure 1 provides valuable insights into the internal mechanics of language models.
- LiLaVe requires only approximately 5k training examples compared to 250k+ for LLM-based verifiers, making it highly practical for data-scarce scenarios and quick adaptation to new datasets.
- The novel conditional majority voting and conditional self-correction strategies demonstrate practical utility and achieve better accuracy-efficiency tradeoffs than baselines, as shown in Figures 4-5.
- Training on CPU in under 15 minutes and achieving 350× inference speedup compared to LLM-based verifiers provides significant practical advantages for deployment.
- The paper demonstrates good transferability across different base models (Llama, Gemma, Phi) as shown in Table 2, suggesting the approach is not overly model-specific.

**Weaknesses:**

- The core novelty is limited as probing hidden states for information extraction is well-established, and Bui et al. (2025) presents a similar approach for code generation, making this work somewhat incremental.
- The comparison with process reward models is insufficient, appearing only in Appendix B.10 with mixed results where Math-Shepherd sometimes outperforms LiLaVe on GSM-style tasks.
- Several design choices appear ad-hoc despite ablation studies: the specific temperature mixture (0, 0.25, 0.5, 0.75, 1.0), the choice of layers (-1, -2, -4, -8, -16), and tokens (-1 through -16) lack strong theoretical justification.
- The paper lacks analysis of failure modes or deeper investigation into what patterns the XGBoost model learns from hidden states and why these representations contain correctness information.
- Cross-dataset transfer results in Table 4 show significant performance degradation (e.g., MATH-trained verifier achieving only 0.53 AUC on algebra_linear_1d), limiting claims about generalization.
- The verifier-oracle gap shown in Figure 10 reveals substantial room for improvement, with oracle selection achieving much higher accuracy than any verifier-based strategy.
- Some experimental choices lack justification: why these specific four datasets, why these particular numbers of votes in majority voting experiments, and how sensitive is performance to hyperparameter variations.
- The MATH results with Gemma 2 2B (AUC 0.53 in Table 2) suggest the approach may struggle when the base model has very low accuracy (approximately 5%), indicating potential limitations.

**Questions:**

1. Can you provide more analysis of what the XGBoost model learns? For example, feature importance analysis or investigation of which hidden state dimensions are most predictive of correctness?
2. How does LiLaVe performance scale with base model size? Would the same approach work with larger models (70B+) or smaller models (1B-3B)?
3. In the conditional self-correction experiments (Figure 4), some datasets show minimal improvement. Can you explain why conditional self-correction works well for algebra_linear_1d but provides limited benefit for MATH?
4. The cross-dataset transfer results (Table 4) show significant drops. Have you investigated why MATH-trained verifiers transfer poorly, and could ensemble approaches improve transfer?
5. How does the computational overhead of extracting and storing hidden states during generation compare to the verification speedup? What is the end-to-end latency comparison?

---

> ### Author Response · Authors · 2025-12-03
>
> **Re: Limited novelty**
>
> Extracting information from hidden states isn't a completely new idea, and we admit that our work was inspired by other publications related to probing. However, we do make original, novel contributions. The most important, original message of the paper is that **extracting latent information from LLMs is practically feasible** and in conjunction with meta-generation strategies it may **significantly boost the performance of the base LLM**.
>
> More specific, novel research items include:
>
> 1. Information from the latent space can be extracted with very robust and fast ML algorithm -- gradient boosted trees. We didn't see this in any other paper related to probing, where the signal is extracted via more involved and less robust procedures.
> 2. We haven't seen any other work analyzing the signal quality depending on the location in the hidden space, as well as the temperature of generations.
> 3. To the best of our knowledge, we show for the first time that self-correction approach can improve the average performance of the LLM when relying on the extracted hidden uncertainty signal.
>
> ---
>
> **Re: Insufficient comparison with PRMs**
>
> We agree with the reviewer that our comparison with process reward models (PRMs) is limited. The experiment in Appendix B.10 is included only for contextualization, illustrating how one strong representative PRM, MathShepherd, performs when used as an outcome reward model (ORM).
>
> Our work does not aim to develop or analyze a PRM, and LiLaVe is neither trained nor evaluated in a PRM setting. To avoid comparing apples to oranges, we deliberately kept this experiment out of the main paper and refrained from drawing strong conclusions from it.
>
> A systematic comparison between latent verifiers and PRMs would require training LiLaVe explicitly as a PRM and evaluating as such. We agree that such an investigation would be valuable, and we plan to explore this direction in future work.
>
> ---
>
> **Re: Lack of analysis of XGBoost predictions**
>
> It is an interesting remark, however, interpreting XGBoost predictions via means of feature importance measure is not meaningful: the input to the model is a real vector of dim. 2048 (or larger) whose particular positions are impossible to interpret.
>
> ---
>
> **Re: Large verifier-oracle gap**
>
> The gap is large, but this is true of LLM-based ORMs as well! It only emphasizes the importance of investigating verification approaches. See the discussion of the similar remark of Reviewer 8DqU.
>
> ---
>
> **Re: How does LiLaVe performance scale with base model size?**
>
> In the Appendix of the paper we include additional models of different sizes: Gemma 2 2B, and Phi-3.5-mini 4B. As can be seen in Table 2, the AUC performance for these smaller models remain very strong, similar to Llama 3.1 8B.
>
> ---
>
> **Re: Can you explain why conditional self-correction works well for algebra_linear_1d but provides limited benefit for MATH?**
>
> We hypothesize that many problems in MATH may simply be too difficult for Llama 3.1 8B and this model struggles to fix its responses if given a second chance.
>
> ---
>
> **Re: Have you investigated why MATH-trained verifiers transfer poorly, and could ensemble approaches improve transfer?**
>
> We don’t know a definitive answer to this. However, in now extended Table 4, we include results for LiLaVe models trained on different combinations of data. It can be seen that MATH-trained LiLaVe transfers poorly to algebra_linear_1d, but when GSM8K training data is added o the mix, the performance jumps (0.53 -> 0.81).
>
> ---
>
> **Re: End-to-end latency measurement**
>
> We didn’t run end-to-end latency analysis of the whole pipeline because our setup from the purely engineering perspective is definitely sub-optimal. For instance, we extract hidden states using Hugging Face framework, which by itself is much slower than, say vLLM – but the latter does not have a convenient option for extracting hidden states during the inference. But with some engineering work this could be vastly optimized, and then the end-to-end latency measurement of the whole LiLaVe pipeline would be meaningful and useful.

---

### Official Review · Reviewer_ZZw5 · 2025-10-28

**Soundness:** 4
**Presentation:** 4
**Contribution:** 4
**Rating:** 6
**Confidence:** 4

**Summary:**

This paper proposes learning a simple lightweight verifier from LLM hidden weights, to guide inference-time scaling methods. The method, LiLaVe, is shown to be reasonably well-aligned with oracle correctness, and boosts performance in various reasoning-intensive benchmarks for some small-mid-size open-source language models (2-8B). The authors detail the impact of different inference parameters on LiLaVe.

**Strengths:**

Overall: The paper dives into the verification and test-time scaling, topics of particular interest today. Instead of token-level verifier methods, the authors work with hidden states, using methods more aligned with those from the mechanistic interpretability communities. The method is well-motivated, experiments well-executed, and results well-detailed. I think that people in the research community would find this an enjoyable interesting read.

## Thorough experimentation

The experimentation with different design decisions, from architecture to inference parameters, makes sense and is reasonably well-documented.

## Well-done analysis offers generalizable insights

In particular section 3 is quite strong. I appreciate that it offers interesting signal outside of latent verifiers itself, such as the role of temperature in simple test-time scaling methods and the interaction between sample budget and score threshold.

**Weaknesses:**

## Impact of generation length?

One might wonder whether correctness signal is largely correlated with something as simple as reasoning length. LiLaVe uses fixed-relative indices to extract signal for accuracy (i.e. -1, -2, -4, -8...)-- it is possible that the latent verifier largely extracts something as simple as absolute position from the hidden state, and uses this to inform the prediction. It would be nice to see some comparisons between LiLaVe prediction and reasoning length. I am also curious how, given its simplicity, LiLaVe performs as a verifier when tested on reasoning lengths outside that observed in its training.


## Specificity of the method to the model and task

It appears that LiLaVe demands a verifier to be trained for each combination of model and task. While each learned verifier is relatively lightweight, this could present a barrier to scalability. It would be interesting to test more general latent verifiers across a family of similar tasks for a given model. Or, at the least, show how the existing trained verifiers perform when tested on related but different tasks.

**Questions:**

* nit: L049 "Wu et al Wu et al"
* What were the lengths of the reasoning chains across tasks?
* Did you observe "a-ha" moments in the reasoning chains used in this experiment, where the chain-of-thought transitions away from confusion or incorrectness? If so, how does the LiLaVe prediction change before and after those? i.e. is there correlation between token-level appearance of correctness and correctness predicted from hidden state?
* Do LiLaVe verifiers generalize across tasks? In the appendix it is noted that the Phi-2B model struggled in the MATH domain where the model may not have generated enough correct generations to train LiLaVe. What happens if the GSM8k Phi LiLaVe verifier was used in the MATH domain?

---

> ### Author Response · Authors · 2025-12-03
>
> **Re: Impact of generation length**
>
> That's an interesting question! We run an experiment to see if LiLaVe scores are correlated with the length of generations, and the conclusion is negative -- the correlation is very low. However, the following is true: whenever the generation length is significantly longer than average generation length for a given benchmark, it always implies the presence of hallucinations, which in turn *almost* always implies incorrect answer. However, the number of hallucinated generations for Llama 3.1 8B remains very low (< 1% across all tested benchmarks).
>
> Regarding the question about the lengths of the reasoning chains across tasks (in tokens):
>
> * algebra_linear_1d: 185.29
> * GSM-Symbolic: 134.75
> * GSM-Symbolic-p2: 297.09
> * MATH: 328.04
>
> These samples were generated with temperature 1. (As a side note: we observed a positive correlation between the temperature and the generation length.)
>
> ---
>
> **Re: Specificity of the method to the model and task**
>
> It is true that the best results are obtained when training a LiLaVe module for a given combination of a task and model. The strength of our approach is the easiness of training such LiLaVe modules -- 1000 or fewer labeled examples and a few CPU-minutes are sufficient to obtain a LiLaVe model which can practically boost the performance of the base LLM. This is samewhat akin to the LoRA technique, where the goal is not to train a general model, but rather adjusting the model to specific data in compute-constrained regime.
>
> However, *LiLaVe does show substantial transfer of performance across tasks*.  We included transfer experiments already at the time of submission in Table 4.  Now, we extended this table with additional training / testing combinations, including a LiLaVe model trained on all considered benchmarks. The main message of this table is that
> 1. LiLaVe modules trained on specific data always perform best on this data.
> 2. The general LiLaVe module trained on all the dataset has slightly weaker performance across datasets compared to specific LiLaVe models.
> 3. The transfer across datasets differs: most of the time is very good (for instance, algebra_linear_1d: 0.93 vs algebra_linear_1d --> GSM8K: 0.86) but sometimes is poor (MATH --> algebra_linear_1d: 0.53).
>
> ---
>
> **Re: "a-ha" moments in the reasoning chains**
>
> We did not observe a-ha moments in the model used in our experiments, Llama 3.1 8B. This model does not fall under a category of "reasoning models", and its chain-of-thought trajectories do not display transitions from generation to verification or any backtracking behavior.
>
> If we were to switch to another model such as Qwen3 4B, we would observe a-ha moments occur at various stages of generation and usually correspond to the correction of particular reasoning steps. This makes the reviewer's question difficult to address within our current setup: LiLaVe is trained to evaluate uncertainty at the level of full solutions, not at the token-level or at the reasoning step-level. In other words, LiLaVe does not currently function as a process reward model (PRM).
>
> We agree that examining how reasoning models internally estimate the likelihood of final success immediately before a-ha transitions is an interesting research direction. However, studying such effects would require extending LiLaVe so that it can ingest and evaluate hidden states from any point in the chain of thought. Developing and benchmarking such a generalized PRM-like variant lies beyond the scope of the present work.
>
> ---
>
> **Re: Do LiLaVe verifiers generalize across tasks?**
>
> To significant extent they do! But this is dataset dependent – see our discussion of similar question of Reviewer 5yvv and a new Table 4 in the paper.

---

### Official Review · Reviewer_8DqU · 2025-11-03

**Soundness:** 3
**Presentation:** 4
**Contribution:** 2
**Rating:** 2
**Confidence:** 4

**Summary:**

Typical verifiers are LLMs that are trained to assess the correctness of responses. However, these can be very expensive to train and deploy, especially in scenarios where one needs to verify many responses (e.g., scaling test-time compute through repeated sampling). This paper proposes an alternative approach where instead of post-training an LLM to verify, they simply train a dataset-specific classifier using the hidden states of the base LLM. This lightweight approach, LiLaVe, obtains high AUC in terms of its predictions on whether a response is correct or not and can be used in various strategies such as best-of-n and weighted majority vote.

**Strengths:**

Originality:
- Using internal LM representations to train a verifier is a conceptually simple and underexplored idea.

Quality:
- Detailed justification for the method is provided (e.g., ablations across token index, layer depth, and temperature)
- The appendix provides many extra details and results, such as comparison to oracle selection, transfer experiments, and using lilave on responses from different models (which allows LiLaVe to apply to closed-source models).

Clarity
- Paper is generally well-written and ideas are presented clearly.

Significance
- The method could have practical value in OOD, personalized, or domain-specific settings where off-the-shelf verifiers may struggle.

**Weaknesses:**

Quality
- Would be interesting to also train LiLaVe using either the base model's embeddings or BERT embeddings.
- Some key baselines are lacking. While Table 1 compares AUC to existing reward models, the BoN and weighted MV performance of these reward models is missing. I was expecting to see Figure 3 also have curves for ORM-Mistral (BoN and WMV) and ORM-Deepseek (BoN and WMV).
- My main concern is that the gains are modest overall and fade at larger test-time scales; most improvements appear only in low-sample regimes. The gap between LiLaVe and oracle verifier remains large.
- Moreover, the poor asymptotic performance of BoN as N increases raises concerns about LiLaVe. In particular, I wonder if high AUC/a classifier is the best thing to learn, since selecting among N may be more of a process that requires high precision and ranking ability.

Significance
- Despite being well-executed, the empirical impact is modest and highly dataset-specific. The method’s dependence on labeled data and limited scaling make it less practical for large-scale deployment. In general settings, I suspect that off-the-shelf verifiers still obtain the best accuracy (although I agree at a higher cost too).

**Questions:**

1. How would an embedding-based classifier compare to LiLaVe?
2. What are the Best-of-N and Weighted Majority Vote curves like for reward models like ORM-Mistral and ORM-Deepseek?

---

> ### Author Response · Authors · 2025-12-03
>
> **Re: Training LiLaVe on BERT embeddings**
>
> LiLaVe is intentionally trained on embeddings from autoregressive LLMs, so that it is usable in meta-generation strategies without much additional computational overhead. We find that the generator’s own embeddings provide sufficiently rich uncertainty signals for this purpose. While we agree that studying embeddings from encoder-only transformers is a curious topic,  it falls outside the scope of our work. It is also against the spirit of the lightweightness of LiLaVe, as is it would additionally require to run GPU-intense inference with large embedding model.
>
> ---
>
> **Re: Comparison with ORMs in BoN and WMV experiments**
>
> Regarding ORMs it is important to notice that their scope is different that the one of LiLaVe. Concretely, they requires orders of magnitude more date to train, and are much more costly in inference.
>
> Still, noting that it is not apple-to-apple, we agree that it is still curious to see the comparison. In the original submission we already compare against ORMs in terms of AUC (Table 1) which, to our pleasant surprise, looks quite favourable for LiLaVe, and despite the aforementioned disadvantages it nearly matches ORMs.
>
> Now, per the Reviewer ‘s suggestion, we additionally compared LiLaVe in the context of BoN and WMV strategies. The results are now included in Figure 11 in the paper.
>
> It can be seen that:
> 1. LiLaVe gives almost identical gains on the algebra_linear_1d dataset.
> 2. LiLaVe is a bit worse on GSM-Symbolic and GSM-Symbolic-p2 benchmarks.
> 3. LiLaVe allows to achieve much higher accuracy on the MATH benchmark.
>
> These results are not completely aligned with the AUC scores in Table 1. For instance, LiLaVe achieves a bit higher AUC on GSM-Symbolic-p2 compared to the ORMs, but behaves worse with BoN and WMV. One possible reason for this may be that in Table 1 we use temperature 0, whereas in BoN / WMV experiments temperature is 1.
>
> ---
>
> **Re: Modest gains with LiLaVe and large gap between LiLaVe and the oracle**
>
> It is indeed true that the gap between the oracle and LiLaVe remains large. However, note that the same is true for the ORMs! (See Figure 11 we describe above.) This only emphasizes the potential of verification and the need for investigating new verification approaches – LiLaVe is  a novel step into this direction.
>
> Also, all metageneration strategies that use oracle remain a very far-fetched upper bound, and not a feasible goal to chase. BoN strategy with oracle corresponds directly to pass@n performance.
>
> We conclude that shrinking the gap is a very interesting research and relevant research project (see for instance [1], where the idea is to ensemble multiple verifiers).
>
> [1] Saad-Falcon et al., Weaver: Shrinking the Generation-Verification Gap by Scaling Compute for Verification, NeurIPS 2025 (https://openreview.net/forum?id=dRjt4vlYVQ)
>
> ---
>
> **Re: Poor asymptotic performance of BoN**
>
> We agree that modest increase of BoN success ratio with increasing n looks worrying at first. After more careful analysis (below), we argue that the asymptotic performance of BoN is a poor proxy of the quality of the verifier.
>
> Namely, it is easy to show an almost-perfect (i.e. very good!) verifier, which results in down-spiraling behaviour of BoN. Essentially, this stems from the fact that even the smallest and least probable failure mode of a verifier compounds and becomes near certain to occur at least once in a large enough sample.
> Consider the following example.
> Let $X^+$ and $X^-$ denote the verifier’s scores on a random positive and negative example, respectively, and let $r \in (0, 1)$ be the probability that the generator produces a correct answer. Define $q := \Pr(X^-  > 2)$. Assume that $\Pr (X^+ \in [1, 2]) = 1$, and $\mathbb{P}(X^-  < 1)  = 1-q$.
>
> Under these assumptions, no matter how small $q$ is (corresponding to an arbitrarily strong verifier in terms of AUROC or similar metrics), the success probability of a BoN strategy is
>
> $(1 - q(1-r))^n,$
>
> which approaches 0 as $n \to \infty$.
>
> This illustrates that it is not reasonable to expect BoN performance to systematically improve with larger n, even when using a near-perfect verifier. In our work, BoN therefore serves only as a baseline. Nevertheless, we observe that when using LiLaVe as the verifier, BoN accuracy does increase, despite the above counterexample.

---

> ### Author Response · Authors · 2025-12-03
>
> **Re: Is AUC the right metric to learn and does it reflect downstream task performance using meta-strategies?**
>
> Thank you for this insightful question. Meta-generation is a multifaceted problem, made even more difficult by the fact that different properties of verifiers may guarantee success of different strategies. Nevertheless, AUC is a very useful proxy. As we show in the following theoretical analysis it offers a lower bound for the performance of conditional regeneration strategy.
>
> Let’s consider a simplification of conditional self-correction strategy – conditional regenerate, which is almost identical, except the second call to generator is identical to the first call and independent (a necessary assumption to reason probabilistically about such behaviour, but does not fully capture the emergent self-correction capabilities of LLMs) then it can be proved that
>
> **Lemma 1:** For threshold $s \in R$, the probability of getting the correct answer via the conditional regeneration method is: $r^2 + r (1-r) (TPR(s) - FPR(s) + 1)$, where $r$ is the probability of generator returning a correct answer for a fixed prompt.
>
> It can be also proved that
>
> **Lemma 2:** Let $x$ be a fixed prompt, and $v$ a verifier with AUC $u \in [0.5, 1]$. Then there exists a threshold $s \in \mathbb R$, such that $TPR - FPR$ of $f(y) = \mathbf{1}[ v(x, y) > s ]$ is at least $1 - \sqrt{2(1-u)}$.
>
> Combining the two lemmas, we get that pushing the AUC of the verifier closes the lower bound of conditional regeneration performance. In particular, when the AUC of the verifier approaches 1, so does the accuracy of conditional regeneration.
>
> We are also aware however that
>
> **Lemma 3:** Let $v_1, v_2$ be verifiers and $AUC(v_1) > AUC(v_2)$. Denote $s_n(v_i)$ the success ratio of best-of-$n$ strategy that uses verifier $v_i$. Then $s_2(v_1) > s_2(v_2)$, but it is not necessarily true that $s_k(v_1) > s_k(v_2)$ for $k = 3, 4, …$.
>
> So various metageneration strategies rely on widely different properties of verifiers. In this work we do not aim to provide a definite answer to what influences that, or which verifier is the best for each metageneration strategy. We provide a lightweight method that can be used to outperform widely adopted metageneration strategies such as majority voting, without much computational overhead.
>
> ---
>
> **Re: LiLaVe’s empirical impact is small and data-specific**
>
> Concrete LiLaVe models may be data-specific, but the methodology as a whole is general and we show its empirical impact is significant! LiLaVe models are not one-fits-all, but rather cheap and specialised to concrete situation and possible scarce data.
>
> One could imagine that it is similar to LoRA in language modeling – having a handful of data we can twist towards a desired behaviour, whereas training LiLaVe is actually much easier than training LoRA (only a few minutes of training on a laptop’s CPU, robust training algorithm, almost instant evaluation once the access to the hidden states is given).

---

### Official Review · Reviewer_5yvv · 2025-11-03

**Soundness:** 2
**Presentation:** 3
**Contribution:** 2
**Rating:** 4
**Confidence:** 2

**Summary:**

The paper proposes LiLaVe, a light-weight logits-based classifier for math verification. First, the paper analyzes the position and layer-dependence of the hidden-states. Then it trains a classifier to verify LLM solutions of math reasoning tasks. Lastly, the trained classifier is applied in adaptive majority voting and self-correction to improve the answer quality.

**Strengths:**

the paper is well-written, and analysis is comprehensive.

**Weaknesses:**

* authors should add analysis on other reasoning datasets, e.g. AIME, AMC, GPQA-diamond etc. and perform in-distribution out-of-distribution analysis, i.e. adding part of the new data into training or not, and then compare the verification accuracy of different variants. this is to check the practicality of the proposed approach.

* accuracy-efficiency analysis is also important to better understand lilave's implications on test-time-scaling. figure 1 in ref [1] is a good example. authors should compare the efficiency between conditional majority voting, self-correction and conventional majority voting approach; and also compare verifier-based and verifier-free methods, since [1] mentioned that SC (majority voting) may be more efficient without verifiers in realistic inference budgets. this analysis should be done on a wide range of datasets, MATH500, GSM8K, AIME, AMC, GPQA-diamond etc.

* more advanced classifier-based verifiers need to be added and compared. e.g. the sota ones on rewardbench leaderboard[2,3].

* line 399 mentioned the tuning of tau to optimize the verifier performance, but this is less practical in realistic use cases when no ground truths are available.

[1] https://arxiv.org/abs/2504.01005

[2] https://arxiv.org/abs/2403.13787

[3] https://arxiv.org/abs/2506.01937

**Questions:**

* Figure 5 appears confusing to me: how is each dot generated? more explanation will be appreaciated

* line 248: “*On one hand, for reasoningintensive problems, low temperatures typically result in better performance*” why is this true? for example, both Qwen3[1]  and DeepSeek R1[2] models suggest using non-0 sampling temperature

* line 538: “*Llama 3.1 8B achieves on it results*” should be “*Llama 3.1 8B achieves results*”?

---

> ### Author Response · Authors · 2025-12-03
>
> **Re: Testing on more benchmarks would be better**
>
> We definitely agree that – as always – the more benchmarks the better. Following the Reviewer suggestion we tested LiLaVe on two additional datasets: AIME and GPQA-Diamond.
>
> For AIME, we trained on 860 problems from 1983-2021 and tested on 103 problems from 2022-2025. We started with Qwen3 8B as a base model which gives good proportion of positive labels (34% and 65% of accuracy in non-thinking and thinking mode, respectively). The LiLaVe’s AUC results are excellent:
>
> * Qwen3 thinking: 0.97
> * Qwen3 non-thinking: 0.96
>
> When Llama 3.1 8B is used as the base model, the resulting AUC lower: 0.67. The likely main reason is low accuracy of Llama on AIME (~12%) which results in too few positives for training LiLaVe. Also, note that Qwen3 is a strong reasoning model on its own and when it cannot produce a correct answer, it may manifest on the token level, so it is easier for LiLaVe to pick up easy patterns.
>
> As for GPQA-Diamond, we randomly split its 198 problems into 130 training and 68 testing examples. Despite this **very small training dataset** the LiLaVe’s AUC (with Llama) on test is very good: 0.76. This is a very positive result showing that one can train strong LiLaVe verifiers with much fewer examples than the 1000 we use in the main experiments.
> We included these additional results in Table 5 in the paper.
>
> However, we wanted to emphasize that the main goal of our work is *not* to train one general verification model which will universally work on a broad range of dataset (which is what ORMs / PRMs try to achieve). Instead, we want to show a practical, lightweight method which can be easily adapted to specific datasets the user deals with. A good example of an established method of that kind is LoRA, but LiLaVe is arguably even easier to train.
>
> Whenever one has access to about 1000 (or even fewer) examples with a mix of positive and negative labels, it is very easy to train a specialized LiLaVe model (see the scripts in the supp. material), which can then bootstrap the base model’s accuracy by coupling a trained LiLaVe with meta-generation strategies. As we show, this works for many (dataset, base model) pairs.
>
> In the main experiment line we use five reasoning datasets, two very popular ones (to facilitate comparison with other approaches) and three synthetic ones (to rule out widespread leakage probems). We think this is sufficient and follows the standards – for instance, see this positively received ICLR 2026 submission [1], also about verification. The authors experiment with one base model (also Llama 3.1 8B Instruct) and only three datasets (two synthetic ones and GSM8K).
>
> [1] Verifying Chain-of-Thought Reasoning via its Computational Graph ( https://openreview.net/forum?id=CxiNICq0Rr )
>
> ---
>
> **Re: Out-of distribution analysis needed**
>
> We provide additional experiments described below, suggesting that LiLaVe enjoys reasonable out-of-distribution generalization. This was somewhat surprising to us, even more we thank for this suggestion. We note that the scope of LiLaVe is akin to the traditional ML models, being lightweight and easy to train on a new dataset. As empahsized above, we think that LiLaVe’s main strength is its plug&play nature (similarly LoRA adapters), which greatly compensates for its perhaps lower out-of-the-box generality (compared to LLM-based models).
>
> Below we show AUC performance for different combinations of training sets on three testing sets.
>
> training data|algebra_linear_1d|GSM8K|MATH
> -----------|----------|-----|---------
> algebra_linear_1d|0.93|0.75|0.71
> GSM8K|0.87|0.86|0.84
> MATH|0.53|0.72|0.88
> algebra_linear_1d+GSM8K|0.94|0.85|0.84
> GSM8K+MATH|0.81|0.85|0.88
> MATH+algebra_linear_1d|0.93|0.81|0.88
> all|0.93|0.84|0.87
>
> We now include these additional results in Table 4 in the paper.
>
> ---
>
> **Re: Comparison with more advanced classifier-based verifier**
>
> As requested, we compared against the current best model on the leaderboard of  RewardBench [2] -- Skywork-Reward-V2-Llama-3.1-8B. Below are the AUC results on the reasoning benchmarks:
>
> | Benchmark              | Skywork-Reward-V2 | LiLaVe |
> |------------------------|-------------------|--------|
> | GSM8K (test)           | 0.7917            | 0.86   |
> | GSM-Symbolic           | 0.7739            | 0.84   |
> | GSM-Symbolic-p2        | 0.7258            | 0.78   |
> | Algebra Linear (1d)    | 0.7706            | 0.93   |
> | MATH500                | 0.8707            | 0.88   |
>
> As can be seen, surprisingly, Skywork-Reward-V2 is much weaker that LiLaVe at ranking responses (and also weaker than the ORMs we compare against in Table 1 in the paper). The likely explanation is that Skywork-Reward-V2 is a general reward model, whereas LiLaVe (and the ORMs in Table 1) were trained specifically on math data – this emphasizes the strength of more specialized methods in this domain.
>
> [2] https://huggingface.co/spaces/allenai/reward-bench

---

> ### Author Response · Authors · 2025-12-03
>
> **Re: Optimizing threshold in the conditional self-correction strategy is not practical**
>
> We respectfully disagree. First, it is actually cheap to estimate the threshold on just a handful of data (it is just one number!). Second, the need to estimate the threshold triggering a self-correction attempt is true for all verifiers which return continuous scores (which is the case for typical ORMs and PRMs).
>
> ---
>
> **Re: Figure 5 is confusing**
>
> Thank you for this remark – we now improved the caption of the figure!
>
> Figure 5 depicts the conditional majority voting strategy which works as follows: generate one sample with temperature 0; if the LiLaVe score is above threshold s, do nothing more; otherwise, additionally generate n samples with temp. 1 and perform majority voting. Different colors in Fig. 5 refer to different thresholds t, whereas different dots on a line of one color refer to different n. The main message of this figure is that for many combinations of t and n, conditional majority voting achieves better accuracy with lower total sample budget.
>
> ---
>
> **Re: Are low generation temperatures actually better for reasoning problems?**
>
> For Llama 3.1 and the one-sample-per-problem setting, low temperatures are better for reasoning problems – we tested this experimentally in Figure 13. This is also aligned with the results in the literature, for instance see Figure 4 in [3]. The situation gets more interesting when sampling n times and applying majority voting: then for different n and different datasets, different temperature levels are optimal (see Figure 13 again).
>
> As for the long-CoT models like Qwen3 – this is actually a very interesting problem – we are not aware of an authoritative published result about optimal generation temperatures here.
>
> [3] Dhuliawala et al., Adaptive Decoding via Latent Preference Optimization.

---

### Author Response · Authors · 2025-12-03

We wanted to thank the Reviewers for their good work and to summarize the rebuttal.

In general, we are delighted that the reviewers found our idea original (8DqU) and practical (KwTJ, 8DqU), our paper well-written (5yvv, 8DqU, ZZw5), our experiments and analysis comprehensive (5yvv, 8DqU).

The main objective of the project was to demonstrate an interesting possibility of lightweight, practical, and competitive alternative to popular LLM-based verifiers, and it seems our message got across.

The reviewers also raised relevant questions and concerns which we addressed by additional experiments and explanations. We already incorporated them in the manuscript. In particular:

New experiments:

* Comparison of LiLaVe and ORMs in the context of weighted majority voting and best-of-n (8DqU) -- now in Figure 11.
* Extended out-of-distributions / transfer experiments (5yvv) -- now in extended Table 4.
* Testing LiLaVe on additional benchmarks (AIME, GPQA-Diamond) (5yvv) -- now in Table 5.
* Comparison with a top reward model from the RewardBench leaderboard (5yvv).

Some reviewers also asked about experiments which actually already are in the appendix:
* Performance of LiLaVe with models of different sizes than Llama 3.1 8B (KwTJ) -- this is in Table 2 (for Phi and Gemma models).

Additionally, we made some minor text improvements, in particular we improved the description of Figure 5 and removed several typos.

We thank the reviewers again for their effort and discussions!

---

### Meta-Review · Area_Chair_LeaQ · 2025-12-24

**Summary:**

The paper proposes lightweight latent verifiers which are trained by taking inspiration from mechanistic interpretability and extracting information from the latent layers of the base model. The authors demonstrate this approach can work, and can be coupled with standard meta-generation strategies like best-of-N sampling.

Overall, the reviewers applauded and appreciated the simplicity of the approach. On the other hand, concerns were raised about the relatively modest performance gains, which seem to shrink even further as the amount of data or compute increases (e.g. when coupled with BoN strategies). Finally, concerns were also raised that the method needs to be retrained (albeit with only little data) for specific tasks/domains.

**Reviewer Concerns:**

Modest performance gains, shrinking with amount of data or compute or more sophisticated meta-generation strategies. Also concerns about it being specific to datasets/domains/tasks.

**Reviewer Scores:**

I think the scores would've remained largely the same, as there wasn't much contention as to whether the above concerns exist, just how severe they are. As the method isn't conceptually new (strategies relying on latent space have been applied to many tasks involving LLMs), the proof-of-concept needs to be stronger / more convincing.

---

### Decision · Program_Chairs · 2026-01-26

Reject